# Review of the Status and Developments in Seaweed Farming Infrastructure

Robert Maxwell Tullberg *, Huu Phu Nguyen and Chien Ming Wang

School of Civil Engineering, The University of Queensland, St. Lucia, QLD 4072, Australia
* Correspondence: r.tullberg@uq.edu.au

**Abstract:** This study provides an overview of both traditional nearshore seaweed farming infrastructure and more recent developments intended for large scale farming in more exposed coastal waters where nutrient supply may be a limiting factor. The success of multi-species integrated multi-trophic aquaculture (IMTA) methods predominantly in East Asia is a clear low cost path to scaling up seaweed cultivation in the broader world that provides for both synergistic sharing of nutrients and reduction in water eutrophication. A number of innovations intended to adapt farming methods to deeper or more exposed coastal waters and semi-automate cultivation steps promise to maintain the viability of farming in higher labour cost countries. Co-location of IMTA/finfish and seaweed farming with grid-connected offshore renewable energy (primarily offshore wind) shows the greatest synergistic benefits for marine space usage, decarbonisation, and nutrient management. Seaweed growth can be accelerated by cycling farm infrastructure between the near surface and nutrient richer depths or upwelling cooler nutrient rich water to sub-surface seaweed crops. Such systems would inevitably require significant increases in infrastructure complexity and costs, jeopardizing their economic viability. Combinations of seaweed and higher value aquaculture products may improve the viability of such novel systems.

**Keywords:** seaweed; infrastructure; longline; offshore wind; renewable energy; IMTA; aquaculture; mussel; bivalve; salmon; finfish

## 1. Introduction and Global Context

The imperative to substantially expand the world's seaweed aquaculture supply is now well established in published literature and has the strong backing of virtually all global non-government organizations (NGOs) [1]. The expansion of seaweed farming is recognised as one of the best approaches to realising many of the sustainable development goals of the United Nations (SDG 1—no poverty, SDG 2—zero hunger, SDG 3—good health and well-being, SDG 8—decent work and economic growth, SDG 10—reduce inequalities, SDG 12—responsible consumption and production, SDG 13—climate action, SDG 14—life below water) [1–3]. Total seaweed production was estimated at about 2.2 million tonnes in 1969 and rose to more than 35 million tonnes in 2019 [3]. Over that period seaweed production from wild collection was almost unchanged (~1.1 million tonnes), but cultivated seaweed production increased exponentially and accounted for 97% of the total seaweed production in 2019 with a compound annual growth rate of 6.2% between 2000 and 2018 [2]. Currently, seaweed farming is regionally unbalanced with over 97% of cultivated seaweed being produced in seven counties in East Asia with China being the largest producer of seaweed [3]. The expansion of seaweed farming in Oceania, Europe, Africa, and the Americas is an opportunity to bring those same benefits of sustainable "green growth" to the wider world, including Australia [4]. Importantly, this seaweed farming growth requires no arable land, freshwater, or fossil-fuel derived fertilisers, all of which are heavily constrained and carry significant costs [5].

The demand for seaweed products in a low-carbon world is extensive and growing [3,6]. Seaweed as a traditional food source (used in salads, soups, sushi wraps, etc.) provides for a low-calorie diet that supplies vitamins A, B, C, and E, dietary fibre, omega-3 fatty acids, essential amino acids, and has been shown to improve digestive health, reduce the risk of colorectal cancer, and reduce obesity [7]. Seaweed-derived hydrocolloids are used in many food products, such as salad dressings, ice cream, and beverages, and account for about 40% of the global hydrocolloids market [8]. Other uses include fertilisers, cosmetics, nutraceuticals, pharmaceuticals, and the emerging markets for bio-plastics, fabrics, bio-fuels, bio-char, and potentially carbon sequestration. Seaweed also plays an important role in the aquatic ecosystem, providing eutrophication mitigation, shoreline protection, and habitat for aquatic organisms as nursery grounds [3].

In response to concerns about greenhouse gas emissions the beef cattle market is being challenged by changing dietary patterns and the growth of lab-grown meat and vegan substitutes. The inclusion of small quantities of specific seaweed species as feed additives has been shown to substantially reduce enteric methane production in beef and dairy cattle. For example, the inclusion of the red seaweed *Asparagopsis Taxiformis* at a dietary dry matter level of just 0.2% yielded a methane reduction of 98% relative to a controlled beef steer group [9]. In another study, the inclusion of a closely related species *Asparagopsis Armata* in Holstein dairy cattle at a rate of 1% dry matter yielded a methane reduction of 67.2% [10]. Notably, these and other studies on seaweed feed supplementation reported simultaneous improvements in feed efficiency for beef cattle and milk production [11,12].

The overwhelming majority of current seaweed farming production is in nearshore-sheltered and intertidal shallow waters where simple systems combine with low labour costs for their feasibility. However, existing nearshore-sheltered farming practice is facing a number of challenges to its continued growth. Many nearshore-sheltered coastal sites are in direct competition with other uses in marine spatial planning—namely tourism, shipping, and fishing [13]. In many places, particularly in tropical and lower latitudes, rising upper pelagic ocean temperatures are limiting the productivity of seaweed farming [2]. There are also a number of environmental risks in nearshore-sheltered farming including the slowing of water flows in protected areas, and the spread of disease and parasites under some circumstances. Farming offshore or in exposed coastal waters would avoid the trampling and shading of natural macroalgal beds and seagrasses, reduce the incidence of herbivorous fish grazing, fouling, and epiphytic growth, and help to reduce ocean acidification [3,14–16].

Moving seaweed farming systems to coastal exposed waters (typically within a country's exclusive economic zone—EEZ) and beyond would allow for an order of magnitude increase in worldwide seaweed annual production. It has been estimated that whereas the ocean currently provides only 2% of total food by weight if just 10% of all ocean areas were eventually farmed for seafood and seaweed aquaculture it would produce an equivalent yield to that produced by land-based agriculture [17]. Estimates of the growth needed in world food production due to population rise by 2050 range between 70–100% underlining how critical the development of offshore and exposed seaweed cultivation technology is in the coming decades [18,19]. Such a large increase in coastal aquaculture would need to consider a range of biological issues such as interference in aquatic animal migration routes, avoiding aquatic animal infrastructure entanglement, other side effects on pelagic aquatics, and changes in nutrient balances.

There have been several attempts to clarify the marine terminology used in the discussion of seaweed and aquaculture in general. For the purposes of this paper, a recently proposed three-level terminology (shown in Table 1. below) will be adopted as most accurately encompassing the range of marine environments [15]. In practice, there are still some site environments that are not easily classified such as the relatively shallow protected waters inside reef structures that may extend well beyond three nautical miles (NM), such as on Queensland's Great Barrier Reef coastline. These might be classified by the additional term, *Reef Protected*.

**Table 1.** Aquaculture environment definitions.

| Category | Water Depth (m) | Distance to Shore (NM) |
|---|---|---|
| Offshore | ≥50 m | >3 NM |
| | <50 m | >3 NM |
| Nearshore-exposed | ≥50 m | <3 NM |
| Nearshore-sheltered | <50 m | <3 NM |
| *Reef Protected* | <50 m | >3 NM |

Whilst there is extensive published literature on seaweed biology and downstream applications, there is comparatively little work investigating the infrastructure and economics of seaweed farming, which is key to the successful scaling of aquaculture worldwide. Thus, this paper compiles the traditional infrastructure to more recent infrastructure developments used in seaweed farming, as well as highlighting their limitations and cost-effectiveness. In Section 2, traditional seaweed farming methods and infrastructure are presented. Section 3 describes the evolution of offshore and semi-automated infrastructure systems. In Section 4, the infrastructure and economics for co-cultivating seaweed and other species are discussed. Section 5 discusses the co-location of seaweed/other species and offshore renewable energy farms. Section 6 presents the concluding remarks and recommendations for future research studies on the infrastructure and economics of seaweed farming.

## 2. Traditional Farming Methods

### 2.1. Background and Seaweed Species Classification

As with any form of aquaculture the infrastructure and cultivation methods used are highly dependent on the seaweed species and their ideal growing environment. The most commonly farmed species and their uses and basic infrastructure are summarised in Table 2. The simplest systems can often provide inspiration for further development. Examples of bamboo rafts and twine-growing frames, tube nets, simple monolines, buoy-supported longline systems, and staked rope bottom culture are illustrated in Figure 1. The most basic and labour intense systems, such as the bamboo rafts, and staked rope bottom cultures (typically used in the tropics in small-scale farming of red seaweed) rely on manual fragmentation and seedling propagule attachment to seeding ropes [20,21]. Simple tube-net systems are used to give improved resistance to nearshore wave damage.

With a few notable exceptions, such as free-floating *Sargassum* spp., almost all wild-found and farmed seaweed rely on seaweed attaching to some structure through their root-like "holdfast" organ. Given adequate sunlight for photosynthesis, nutrient availability, salinity, temperature range, oxygen, and carbon dioxide, seaweed propagules can grow regardless of how deep the sea beneath is.

Depending on the fragility of the species, the water speed, through wave action, currents, or tides, needs to be kept within upper bounds to avoid damage and lower bounds for optimal nutrient uptake and growth [20,22,23].

It has been shown that increased tension in sub-surface culture lines increases water flow and, hence, the growth of some kelp species in low current and surface wave conditions [22,23]. However, in moderate to higher energy environments though, this increased tension reduces the ability of the culture line to dampen wave and current energy risking seaweed holdfast detachment and higher mooring loads. A further consideration for system design is the buoyancy of different species. Positively buoyant species such as *Macrocystis* spp. that develop pneumatocysts (air-sacs) grow up from their holdfast, whereas most other species are typically slightly negatively buoyant. This variation in species buoyancy, size, and drag inevitably means that infrastructure systems need to be customised and adapted into solutions to suit different groups sharing similar characteristics.

**Table 2.** Seaweed species use and infrastructure.

| Species Group | Seaweed Colour | Biomass Load per Metre (Scale 1–3) | Tonnes (Wet) Annual Cultivation [3] | % World Market [3] | Buoyancy | Region | Applications | Cultivation Method(s) | Hydrodynamic Suitability (Scale 1–3) |
|---|---|---|---|---|---|---|---|---|---|
| Laminaria/Saccharina | Brown | 2—Moderate | 12,273,748 | 35.4% | Neutral/slightly negative | Temperate | Human consumption; Raw material for alginate, mannitol, iodine, Abalone feed | Longlines | 2—Moderate, grows in exposed water |
| Undaria (wakame) | Brown | 2—Moderate | 2,563,582 | 7.4% | Negative | Temperate | Sea mustard, Abalone Feed | Longlines | 1–2—Low–moderate, grows in exposed waters |
| Macrocystis pyrifera | Brown | 3—High | 2 | 0.0% | Positive | Temperate | Food and Cosmetic Products, Animal Feed | Longlines | 2—Moderate, grows in exposed waters |
| Sargassum | Brown | 3—High | 304,000 | 0.9% | Positive | Tropical | Food and Cosmetic Products, Animal Feed | Longlines | 2—Moderate, grows in exposed waters |
| Alaria esculenta | Brown | | 105 | 0.0% | | Temperate | Animal feed | Longlines | 2—Moderate, grows in exposed waters |
| Eckolonia, Lessonia | Brown | 2—Moderate | - | - | Negative | Temperate | Human consumption, fertiliser, animal feed (e.g., livestock, aquaculture), nutraceuticals, and biopolymers and bioplastics | Longlines | 2—Moderate, grows in exposed water |
| Durvillaea | Brown | 3—High | - | - | Negative | Temperate | Alginate industry, fertiliser | Longlines | 3—High, grows in very exposed waters |
| Kappaphycus/Eucheuma | Red | 2—Moderate | 11,622,213 | 33.5% | Negative | Tropical | For carrageenan extraction | Longlines/nets | 1–2—Low–moderate, grows in open water |
| Gracilaria | Red | 1—Low | 3,639,833 | 10.5% | Negative | Tropical | Feed for abalone; For agar extraction; Bioremediation | Longlines/nets | 1–2—Low–moderate, grows in exposed water |
| Porphyra | Red | 1—Low | 2,984,123 | 8.6% | - | Temperate | Food wrap | Nets | 1—Low–delicate |
| Asparagopsis | Red | 1—Low | - | - | - | Tropical/warm temperate | Animal feed | Longlines, net tubes | 1—Low–delicate, unsuitable for open waters |
| All Green | Green | 1—Low | 14,019 | 0.0% | | Tropical/temperate | Human consumption | Land-based facilities | 1—Low–delicate |
| **TOTAL** | | | **34,679,134** | | | | | | |

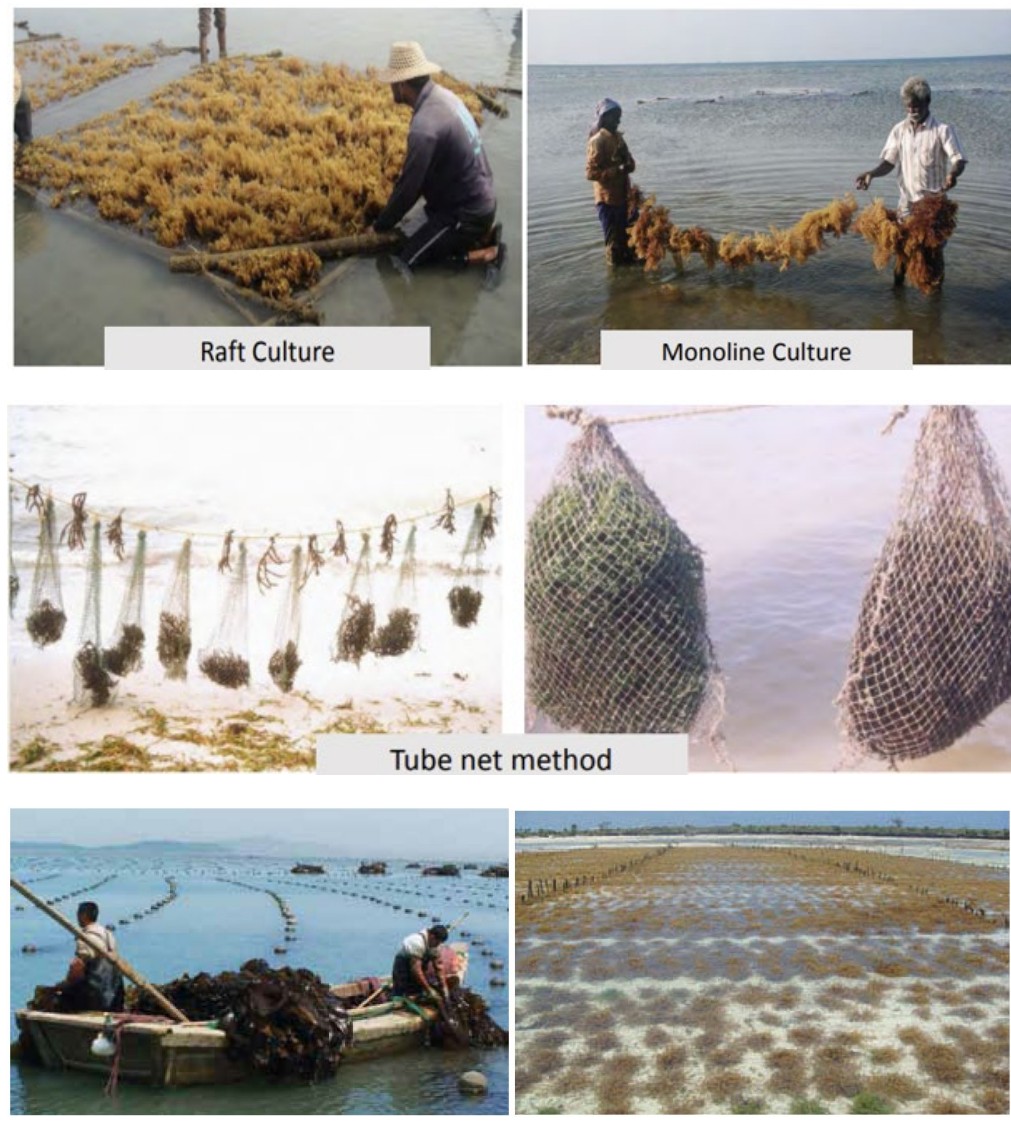

**Figure 1.** Simple seaweed systems—raft, monoline, tube nets-images from https://www.slideshare. net/zoysa89/sea-weed-farmingsouth-east-asia (accessed on 22 September 2022), floating longlines (*bottom left*) [1], staked rope bottom culture (*bottom right*) [1].

## 2.2. Cultivation Systems

Larger operators in more temperate regions typically automate the seeding of longlines in laboratory based hatcheries [24]. By adjusting light and temperature conditions in spawning tanks seaweed spores are induced, settled, and grafted onto coiled longlines. These are subsequently deployed at a suitable depth by using float buoys and concrete moorings in the grow-out phase. Depending on the species, these longlines are either harvested completely and replaced with new freshly seeded longlines or trimmed every few weeks and allowed to regrow multiple times throughout a growing season or for several years in some cases—a technique known as multiple partial harvesting. An alternate variation uses fine strings to seed spores, which are then in turn bound to a larger culture rope, as shown in Figure 2 [22]. Depending on the species and its buoyancy, line systems must be devised that control the average depth of the seaweed for growth optimisation, minimise damage to due wave and current action, and allow for efficient harvesting. The major variations of traditional culture line systems used for most neutral or negatively buoyant species are categorised as horizontal longlines, vertical lines, or garland lines are shown in Figure 3 [19].

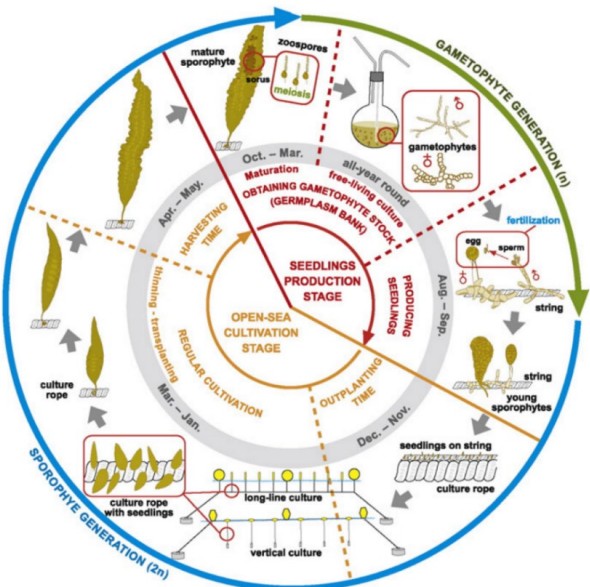

**Figure 2.** Typical cultivation lifecycle in the northern hemisphere of *Saccharina latissimi* using seeded strings. *Reproduced with Permission* [22].

For some species, such as many of the green seaweeds, their weak resistance to waves and currents makes them unsuitable for farming in open waters. These are typically grown in tank-based land facilities where water temperature and water biochemistry can be easily controlled. However, the higher costs inherent in these systems restrict this market to high-value (food uses) applications. Examples of land-based seaweed farming are shown in Figure 4.

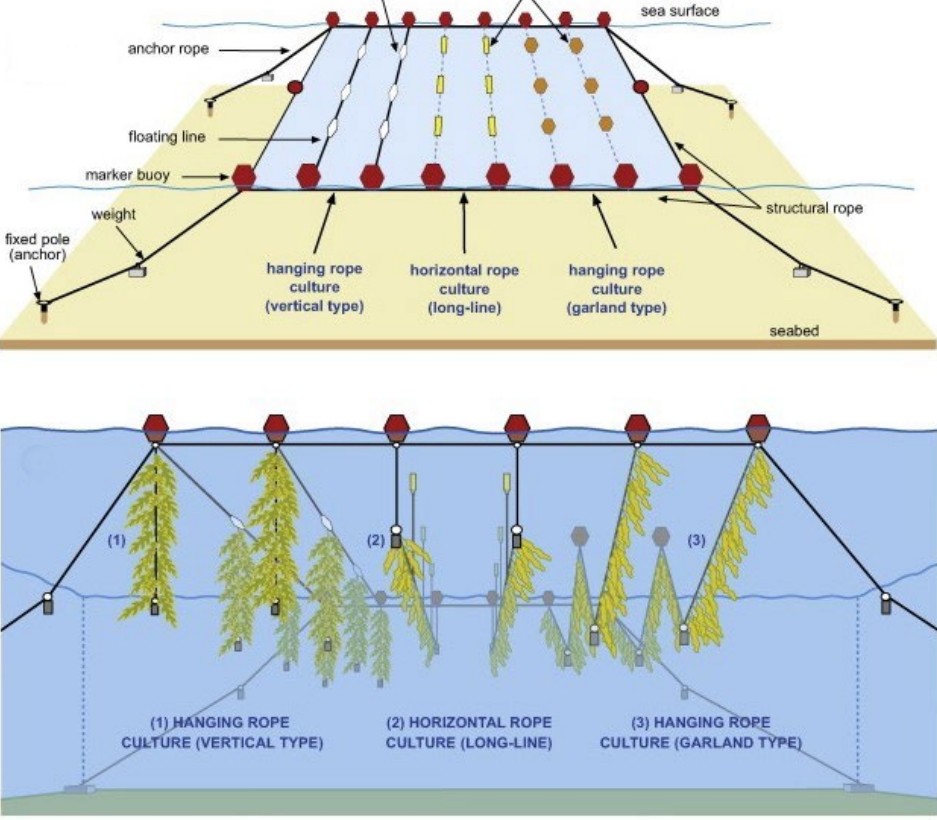

**Figure 3.** Variations on traditional culture rope systems. *Reproduced with Permission* [22].

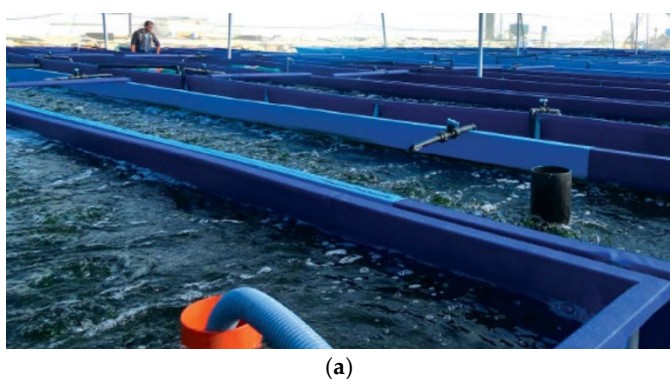 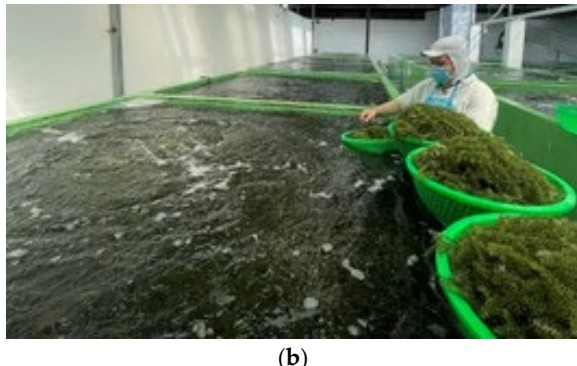

(**a**) (**b**)

**Figure 4.** (**a**) Farming Ulva and Gracilaria in Israel (https://seakura.co.il/en/ (accessed on 22 September 2022)), (**b**) farming Umbibudo sea grapes in Vietnam (http://www.dtvietnam.com/# (accessed on 22 September 2022)).

## 3. Evolution of Offshore and Semi-Automated Systems

In broadening the worldwide reach of seaweed farming, many existing traditional methods can be adapted to new locations where similar socioeconomic and nearshore-sheltered conditions prevail. However, for take-up in the developed world, farming systems must be scaled-up to produce economies of scale and mechanized to overcome higher local labour costs. The development of such systems adapted to offshore and nearshore-exposed production can then be adopted widely. As with any technological development, the systems and modules most likely to succeed will learn and evolve from the fundamentals of the simple systems incorporating step-change advancements where economically feasible.

A study of seaweed farming scaled up to nearly 400 km² in Sweden (a high labour cost country) indicated profitability and positive NPV (net present value) bodes well for future seaweed industry growth in the developed world [5]. On the other hand, a study examining the feasibility of offshore seaweed production in offshore wind farms in the Dutch North Sea in 2016 concluded that average revenues for seaweed products would need to increase by 300% for the seaweed production to break-even [25]. As the authors noted the literature and data sources for seaweed farming costs are scarce and difficult to compare due to variations in species, location, and environmental conditions. Analyses of their cost assumptions indicate an overly pessimistic view. For example, a labour rate (including on-costs) of USD 50/hour is quoted assuming wind farm technicians could also be supplying harvest labour. This compares poorly to a minimum adult hourly labour rate of approximately EUR 10.13 (before on-costs) in the Netherlands in 2022 [26]. The assumed CAPEX (investment cost) for the installation of USD 138,000/ha is 60% larger than the carefully calculated estimate of the BAL system at USD 82,000/ha in 2019 [27]. The BAL system estimates total fixed and variable costs averaging USD 8846/ha/year as compared to USD 18,500/ha/year-roughly double over a 10-year period. The assumed average sale price of USD 555/tonne (dry matter) is relatively conservative and does not allow for growth in the high-value food market. The study assumes annual reseeding of culture lines rather than more profitable multiple partial harvesting over a number of years which is common practice in other regions. Finally, the study assumes the cultivation of seaweed in the absence of other aquaculture crops such as mussels or finfish that would supply nutrients that could lead to yields greater than the conservative assumption of 20 tonnes/ha/year (dry matter).

To facilitate the growth in exposed or offshore seaweed farming cultivation systems must overcome a range of challenges such as:

1. Cultivation structures must be able to withstand infrequent but intense weather events such as storms/cyclones and their associated high-energy waves, and strong currents depending on citing [3,13,28,29]. Whilst the occasional loss of a seaweed crop could be tolerated due to storm action the long-term integrity of structures must be ensured.

2. Offshore waters at the near surface level typically have a lower nutrient density than nearshore waters (with some exceptions) which may reduce seaweed growth rates in the absence of other strategies [13,28].
3. Cultivation infrastructure systems need to be refined to support high productivity, and hence low overall cost harvesting and reseeding operations to be competitive with nearshore farming.
4. Seaweed service vessels for harvesting, reseeding, and transport need to be further developed to suit the in-water infrastructure system and for sharing with other aquaculture and offshore renewables maintenance and repair.

In recent years, several trial and packaged solutions have emerged that attempt to increase the level of automation and modularise the approach to seaweed farming. This paper will focus on the deployment, grow-out, and harvesting operations excluding hatchery technology and post-processing operations such as cleaning, drying, bio-refineries, and conversion to other applications. These solutions are all based on the standard approach of growing seaweed on submerged substrates at a typical depth of 3–10 m beneath the bulk of wave turbulence.

The potential of step-change solutions based on cycling farm substrates between shallow and nutrient rich deep water (depth-cycling) or pumping cool nutrient rich water to the surface (upwelling) requires detailed analysis and is the subject of ongoing research. Recent trials of depth-cycling *Macrocystis pyrifera* to a depth of 80 m off the coast of California showed a fourfold yield increase relative to a control surface crop [30]. Similar trials of cycling kelp to a depth of 150 m in the Camotes sea in the Philippines conducted by The Climate Foundation reported a 300% cumulative growth rate in 45 days relative to a slight negative growth in the surface control crop [31]. The Climate Foundation also conducted trials on upwelling water from a depth of 250 m to a *Kappaphycus* spp. sample crop. The trial crop that was irrigated with cooler nutrient rich upwelled water doubled its biomass over 40 days relative to a surface control sample that lost 25% of its biomass. These early trials demonstrate that technology can be used to overcome the problems of poor growth due to high surface water temperatures and insufficient nutrients. However, early concept analyses of solar or wind-powered implementations suggest the complexity and costs of such systems at scale are likely to present significant economic feasibility hurdles for their adoption when they are deployed solely for seaweed cultivation. Preliminary analysis of these options suggests that depth-cycling arrangements that rely only upon changes to system buoyancy have a specific energy requirement of approximately two orders of magnitude less than upwelling systems depending on a range of factors. Such systems would still have a requirement for (renewably powered) air compression for buoyancy control, depth sensing, and communications to activate submergence under storm conditions. The economic viability of such a depth-cycling system could be significantly improved with the integration of higher value (finfish/seafood) aquaculture.

In developing semi-automated seaweed farming systems, it is critical that the life-cycle carbon emissions of the whole system are considered. A study of Irish seaweed farming based on hatchery seeding and floating longline grow-out operation over a 20-year period that incorporated calculation of infrastructure construction embedded emissions, operating emissions, and carbon sequestration indicated a nett-negative global warming potential of $-0.11$ kg $CO_2$/kg (algae) *wet weight*/year [32]. Given the conservative assumptions in this study on the sources of embedded energy, careful design of more automated solutions should ensure this nett $CO_2$e sequestration is maintained. However, more recent research studies on carbon accounting in seaweed farming have queried some previous assumptions about longer-term carbon sequestration [33,34].

Some generalisations can be made about the different approaches to cultivation systems. In order to hold a certain spatial layout, systems either need to rely on being "pegged-out" under tensile loads supplied by multi-point mooring and floatation buoys, or alternatively need to have internal resistance to compressive loads which necessarily implies a far heavier structure with consequentially higher costs and embedded emissions.

Thus, semi-automated package solutions are most easily classified by their form factor into three generic categories:

1. Linear—advances on the traditional longline systems.
2. Circular—systems typically borrow technology from the seafood farming industry.
3. Two-dimensional (2D)—based on substrates such as fabrics and 2D net structures.

A historical qualitative review of trial cultivation structures (published in 2018), which mainly focused on temperate species, nominated two systems that showed the most potential for further expansion [7]. A summary of that assessment is presented in Table 3. These two linear systems—The MacroAlgal Cultivation Rig (as shown in Figure 5) and The BioArchitecture Lab (BAL) cultivation rig (as shown in Figure 6) both demonstrated realistic economics, technical viability, and successful trials in varying exposed current and wave conditions. The innovative approach of the H-frame spar buoy (as shown in Figure 7) in providing a passive wave-powered mechanism to adjust the depth of a longline array may yet be redeployed in new systems. Similarly, the circular ring concepts pioneered in the German North Sea (as shown in Figure 8) have shown that they are able to grow certain species whilst withstanding current speeds up to 2 m/s and significant wave heights over 6 m [35].

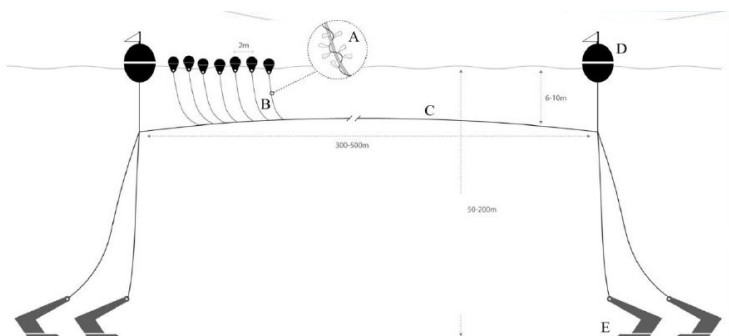

**Figure 5.** MacroAlgal cultivation rig. *Reproduced with permission* [37].

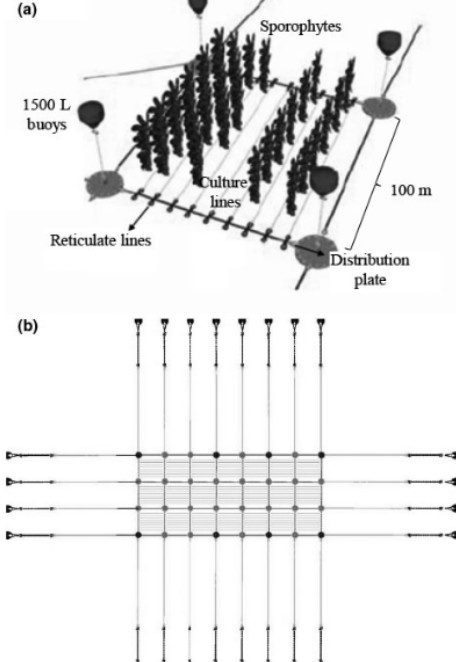

**Figure 6.** BAL system growing positively buoyant *Macrocystis Pyrifera*—(**a**) 3D partial layout, (**b**) plan-view mooring arrangement [27].

**Table 3.** Referenced copy of qualitative assessment of offshore macroalgal cultivation structures. *Reproduced with permission* [15].

| Structure Name/Project Name | Origin Country, Location | Test Period | Site Category | Site Description | | | | Size of Test Area (ha) | Aquaculture Output (Tonnes Ha$^{-1}$ yr$^{-1}$) | Yield (kg m$^{-1}$ rope yr$^{-1}$) | Species | Years Tested at Sea | Technically Viable | In Operation Today | Cost (US$) | References |
|---|---|---|---|---|---|---|---|---|---|---|---|---|---|---|---|---|
| | | | | Distance to Shore (km) | Location Depth (m) | Maximal Sign. Wave Height (m) | Maximal Current Speed (cm s$^{-1}$) | | | | | | | | | |
| Marine Biomess Program | USA (California) | 1970–1983 | N.E. | ~1 | | | | 0.48 | 300 # | | MP | <1 | No | No | CAPEX: 570,000,000 OPEX: 61,400,000/yr. | Harger and Neushul (1983), Neushul (1987), Neushul et al. (1992) |
| BAL's cultivation grid (BAL) | Chile (Quenac, Caldera & Ancud) | 2010–2013 | N.E. | | 60 * | 3 | 115 | 21 | 124 * | *Mean 12.4 * * | MP | 3 | Yes | No | CAPEX: 6000/ha OPEX: 8000/ha | Buschmann et al. (2014), Camus et al. (2018b) |
| Offshore Ring System | Germany (North Sea) | 1995–2002 | O.S. | <5 | 14 * | 6.4 | 152 | | 109 # tonnes dw yr$^{-1}$ | | SL | | Yes | No | | Buck and Buchholz (2004, 2005), Buck et al. (2004) |
| A culture raft | Spain (Matalena) | 2000–2008 | N.S. | ~1 | ~20 * | 3 | 92 | 0.12 | 45.6 # | *Max. 16 * | SL, UP | <1 | | No | | Peteiro et al. (2014, 2016) |
| H-frame structure using SPAR | The Netherlands (Texel) | 2011–2013 | O.S. | 12 | 22 * | 8 | | 0.04 | | | SL, LD | <1 | | No | | Pers. Comm. Hortimare 2018, The North Sea Farm Foundation (2018) |
| Tension-Leg Platform (TLP) | Republic of Korea (Jeju Island) | 2010–2012 | O/N.E. | | | | | 4 | 300 # | *Max. 80.6* | SJ | 2 | Yes | | 0.5 million/ha | Chung et al. (2015) |

**Table 3.** *Cont.*

| Structure Name/Project Name | Origin Country, Location | Test Period | Site Category | Site Description | | | | Size of Test Area (ha) | Aquaculture Output (Tonnes Ha⁻¹ yr⁻¹) | Yield (kg m⁻¹ rope yr⁻¹) | Species | Years Tested at Sea | Technically Viable | In Operation Today | Cost (US$) | References |
|---|---|---|---|---|---|---|---|---|---|---|---|---|---|---|---|---|
| | | | | Distance to Shore (km) | Location Depth (m) | Maximal Sign. Wave Height (m) | Maximal Current Speed (cm s⁻¹) | | | | | | | | | |
| Seaweed Carrier | Norway Trond-heim | 2009 | N.E./N.S. | | | | | | | | SL | | Yes | No | | Seaweed Energy Solutions (2018) |
| MacroAlgal Cultivation Rig (MACR) | The Faroe Islands (Funnings-fjrdur) | 2010 | N.E. | 0.5 | 70 * 200 # | 4 * 6 # | 25 | 9 | 35 * | *Mean* 6 */58 | SL, AE, LD | 8 | Yes | Yes | CAPEX: 13,364/ha OPEX: 10,676/ha | Bak et al. (2018) |

Site categories: O, offshore; O.S., offshore sheltered; N.E., nearshore-exposed; N.S, nearshore-sheltered. All values: * indicates field measurements; # indicates estimates. Species: MP, *Macrocystis pyrifera*; SL, *Saccharina latissima*; SJ, *Saccharina japonica*; LD, *Laminaria digitata*; AE, *Alaria esculenta*; UP, *Undaria pinnatifida*. Cost: CAPEX, capital expenditure; OPEX, operational expenditure.

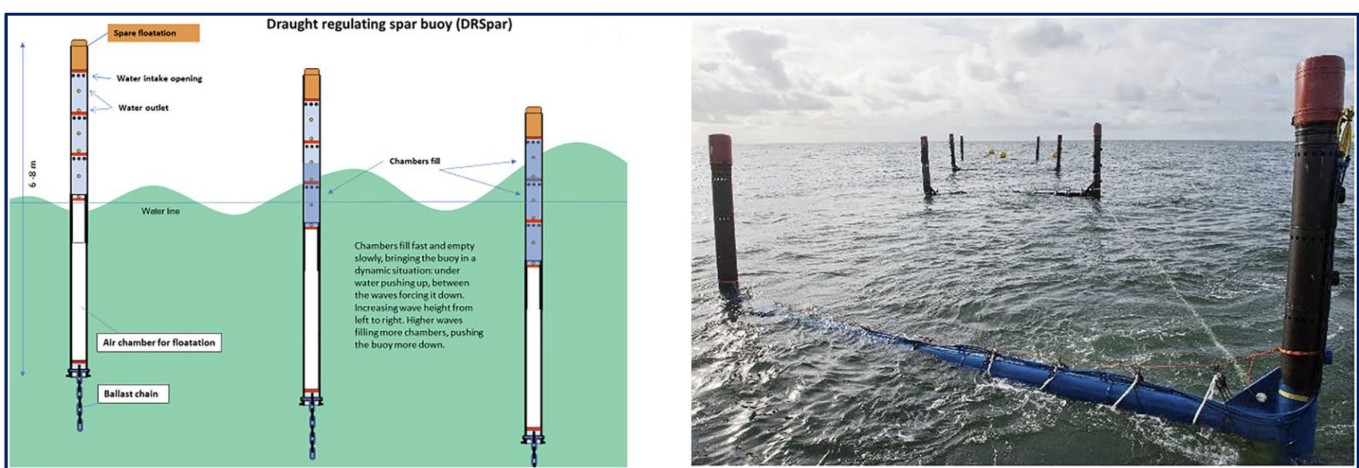

**Figure 7.** H Frame with wave adaptive spar buoy system. *Reproduced with permission* [15].

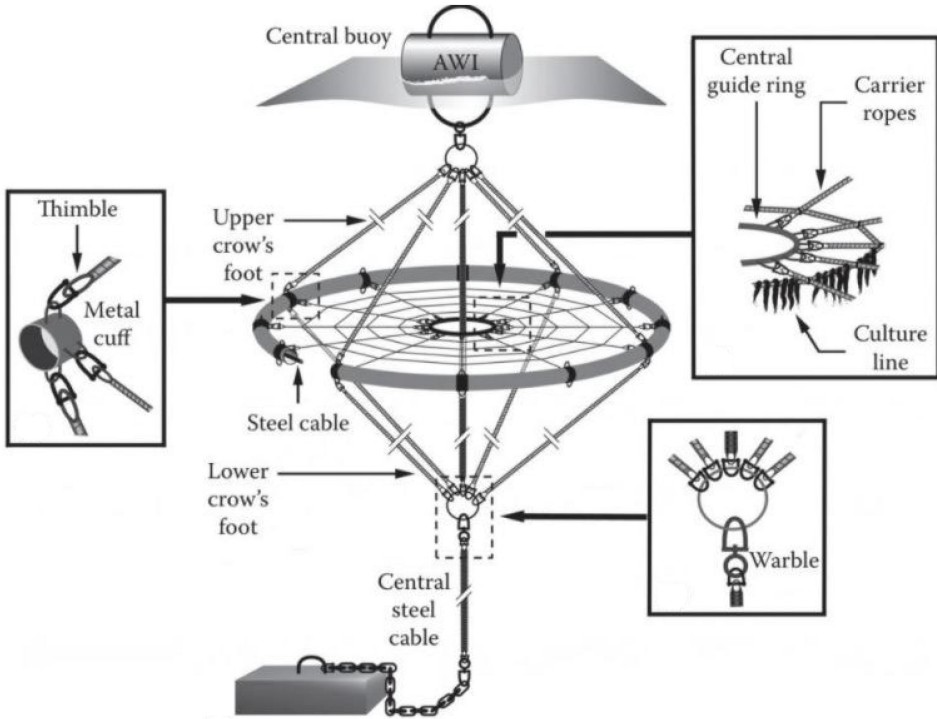

**Figure 8.** Circular sub-surface ring structure [35].

The recent emergence of a similar circular modular system by *Seatech Energy* (As shown in Figure 9) suggests some optimism that such systems may find new markets although their claims to harvest yields and automation are untested.

One potential method of automating the harvest of circular systems incorporates the use of a robotic tool to work around the circumference of each unit harvesting and reseeding lines—a concept labelled *SPOKe* (standardised production of kelp) as shown in Figure 10 [36]. Such a tool could then be deployed from a larger transport vessel to service co-located groups of ring modules. No detailed analysis of this system's feasibility has yet been published.

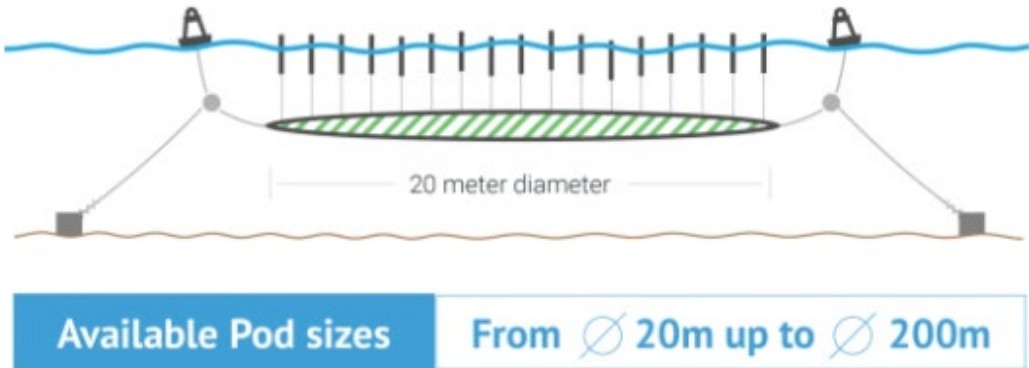

**Figure 9.** Circular modular systems—*Seatech Energy* (https://seatechinnovation.com/seaweed-solutions/ (accessed on 22 September 2022)).

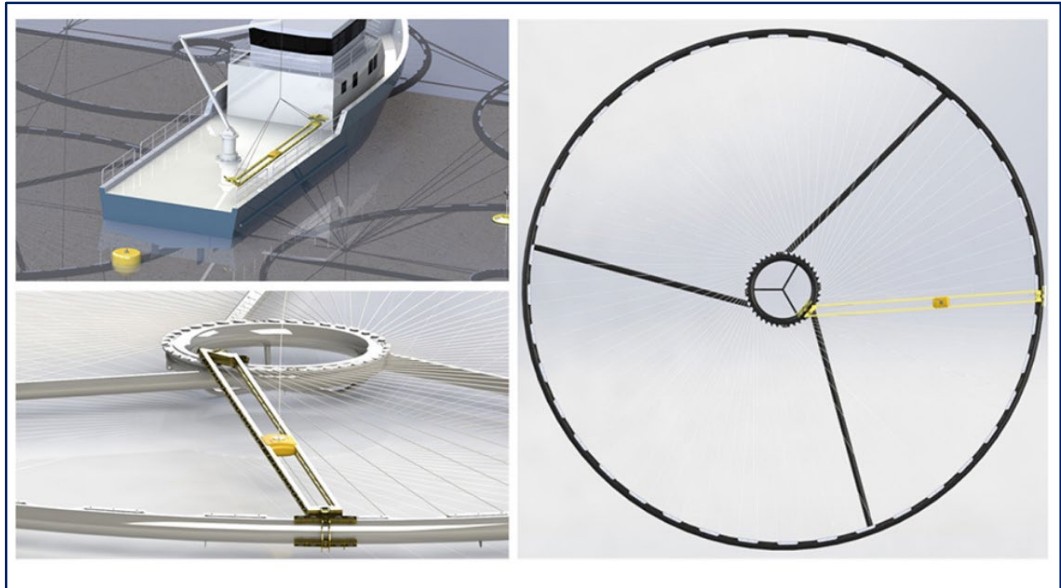

**Figure 10.** SPOKe concept of deployable seeding/harvesting tool [36].

In moving to offshore and exposed waters all systems regardless of form necessarily need to consider the likely maximum biomass loading and consequently the magnitude of mooring load requirements. A cultivation system that is more regularly harvested (either by complete or partial harvest trimming) will lead to lower average thallus or frond lengths, lower biomass per metre, and hence lower drag (Figure 11).

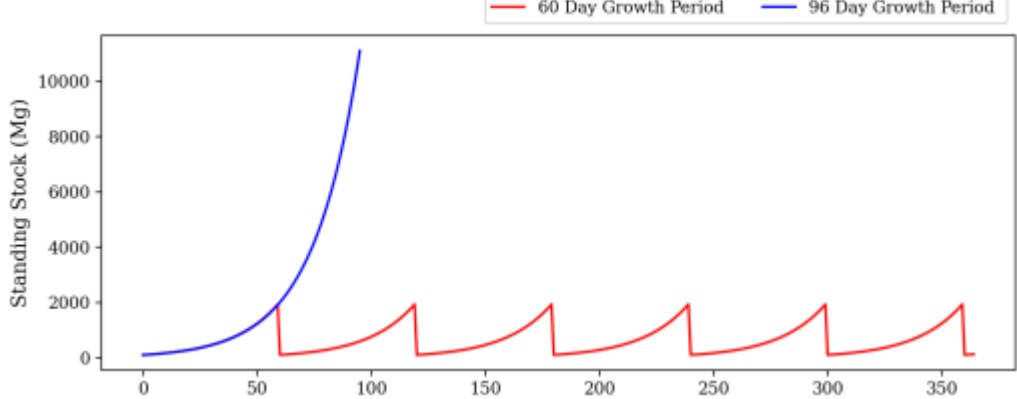

**Figure 11.** Growth Period Trade-off.

An alternative to a multi-point spatial layout (either rectangular or circular) is to use single point mooring and allow a dominant current flow to draw out a linear (or extended rectangular system) under tension provided by seaweed drag allowing for both rotation and submergence [13]. A concept called the *SUBFlex* developed by *GiliOcean Technology* (as shown in Figures 12 and 13) uses this principle. This substantially simplifies mooring but incurs a penalty when the spatial harvest efficiency of the circular orbit is considered. However, when this approach is coupled with an integrated multi-trophic aquaculture (IMTA) system combining both fish and seaweed farming the residual faecal and feed waste not taken up by seaweed is spread over a much larger area. With the added ability to submerge such a linear system upstream nutrients falling to the sea floor can be more easily absorbed by seaweed overnight in a similar depth-cycling arrangement [15]. How such a linear tensile drag model responds in conditions of low currents facing tidal reversals is not clear—it would need to pivot in a stable fashion and not collapse in low drag conditions. The *SUBFlex* implementation appears to use an arrangement of circular modules arranged linearly to provide some degree of compressive or beam strength to the overall system. This concept also relies on the inflation of tubular beam elements to control the buoyancy and depth of the cage system. A similar technology (*Seastrut*^TM developed by *Impact-9*) making use of inflatable and water-filled tubes to control buoyancy can potentially provide for a buoyancy and depth-controlled beam with some bending resistance in a tensile structure that can attenuate wave responses. Crucially, as compared to the ultra-high molecular weight polyethylene (UHMW PE) floating collars typically used in finfish pen aquaculture, these beams would crease rather than fail catastrophically under very high bending moments that may be experienced in high energy ocean environments in storm conditions. These beam elements shown in Figure 14 could well contribute to the development of offshore aquaculture systems both for finfish pens in high energy environments, and combined IMTA seaweed farming.

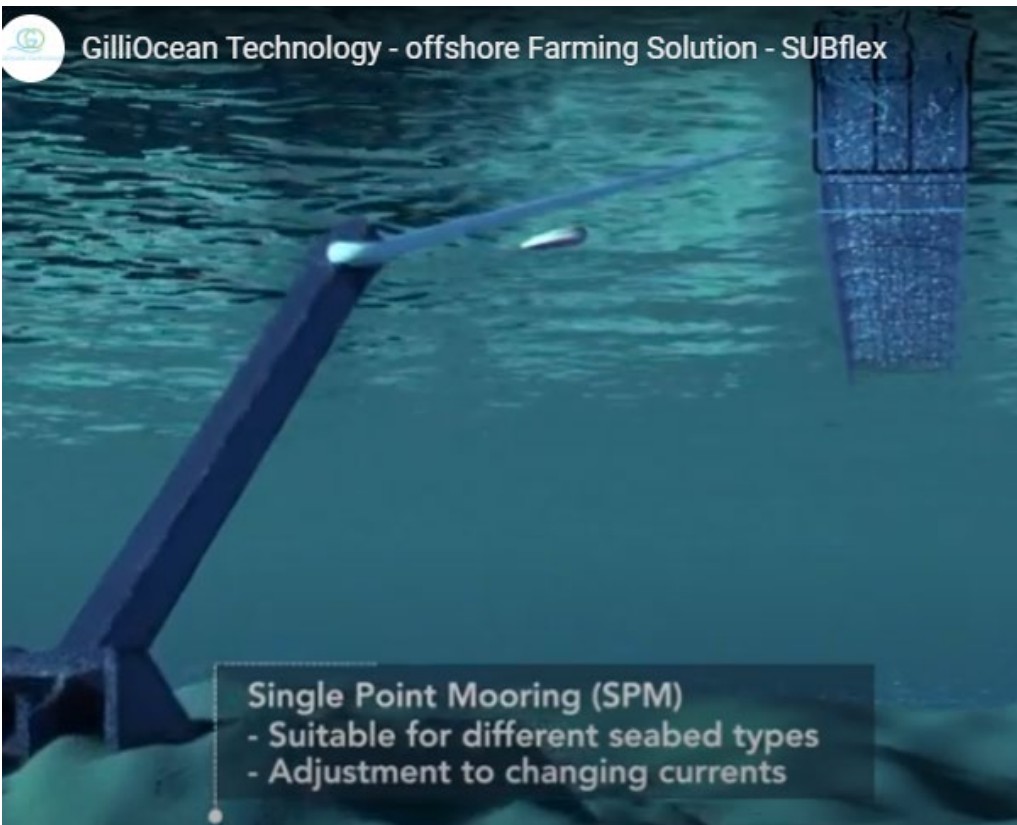

**Figure 12.** Linear tensile system—single point mooring—*Gili SUBFlex*— (https://www.giliocean.com/selected-projects (accessed on 22 September 2022)).

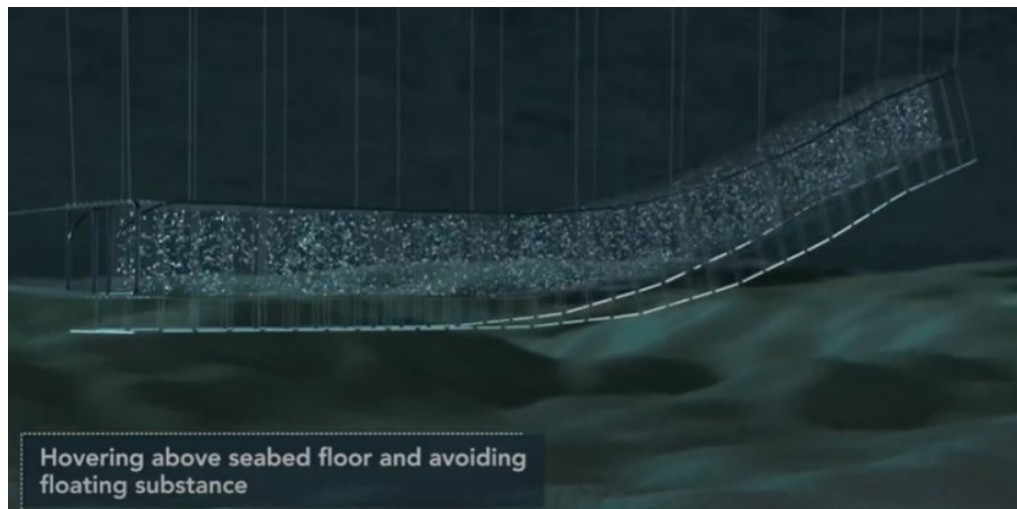

**Figure 13.** Linear tensile system *Gili subflex*–submerged— (https://www.giliocean.com/selected-projects (accessed on 22 September 2022)).

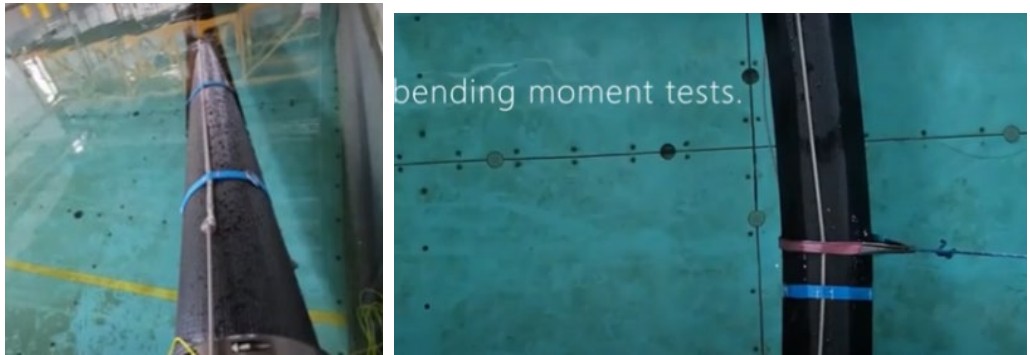

**Figure 14.** *Seastrut^{TM}* under testing—https://impact-9.com/technology (accessed on 22 September 2022).

One of the issues that is reported frequently for linear grid systems is the susceptibility to longline and seaweed entanglements where lines are close and under insufficient tension [7]. This has traditionally been mitigated by the use of some form of perpendicular control rope or other separation devices [38]. This issue can be mitigated by maintaining rope tension in two dimensions more easily through the use of triangulated longline layouts. The *Buland 10* Cultivation system developed in Norway uses this principle to deploy a 2D net as shown in Figure 15.

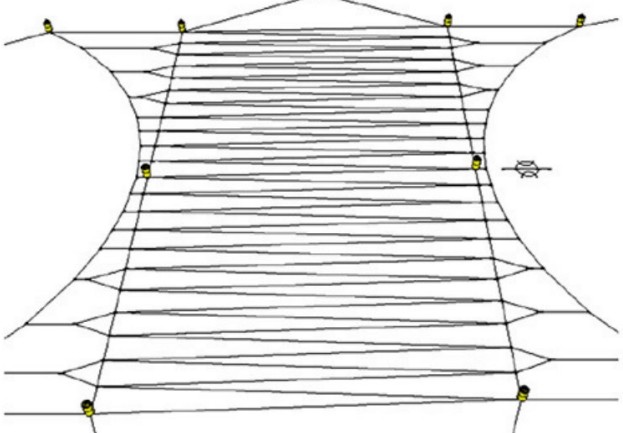

**Figure 15.** *Buland 10* triangulated net long-line system [36].

Although this system may appear to require less mooring and floatation support than the *BAL* grid system the triangulation leads to a considerable reduction in spatial efficiency. In addition, there is little published information comparing the economics of the two systems as yet or discussion of whether entanglement occurs near the longline triangular vertices.

Regardless of mooring arrangements most traditional longline deployments incorporate attached floats and weights (as shown in Figure 4) with the result that such long-lines need to be handled in a hand-over-hand manner (as shown in Figure 16) as opposed to being continuously fed through a linear machine. Although it is difficult to be certain, it is easy to speculate that a long line that could be quickly detached and reattached in a system that allowed for higher speed continuous harvest feed such as the *BAL* system (either complete or trimmed) may offer improved harvest productivity. A follow-up economic study of the BAL system farming *Macrocystis pyrifera* in Southern Chile showed a break-even point of USD 87/wet tonne, as compared to a market price average of USD 470/wet tonne for brown seaweed [3,30]. A linear net system (as shown in Figure 17) that exploits linear automated seeding, and linear-fed shear trimming for multiple partial harvests has been proposed by *Sea6Energy*. Additionally, they envisage a harvest catamaran system that trims and reseeds longlines that in turn transfers harvested seaweed to a larger transport vessel.

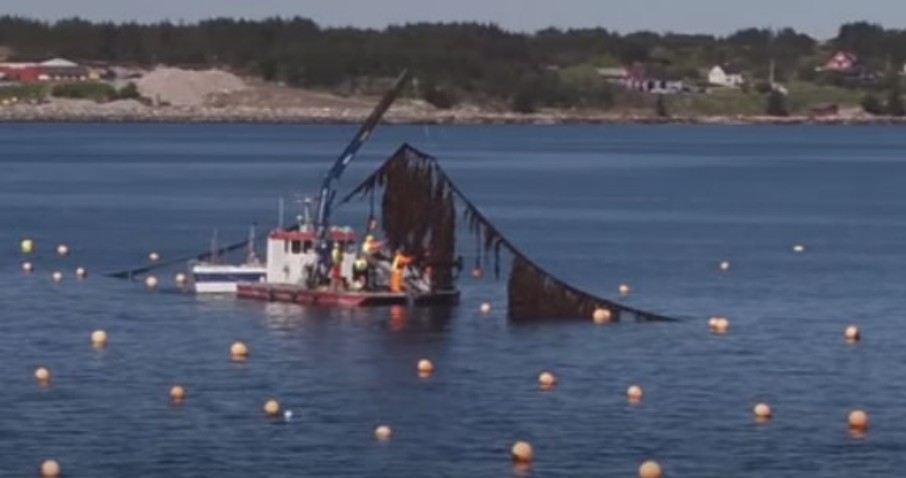

**Figure 16.** Crane manipulating longline with hand-over-hand harvesting—image from https://seaweedsolutions.com/ (accessed on 22 September 2022).

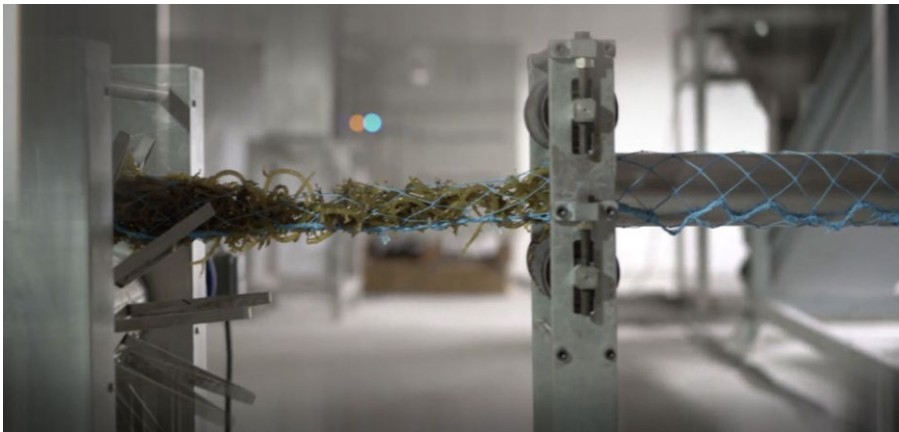

**Figure 17.** Linear net seeding—*Sea6Energy*— (https://www.sea6energy.com/automated-farming (accessed on 22 September 2022)).

The alternative to culture lines either in rectangular or circular/spoke-wise grids is to use a 2D substrate such as a fabric sheet or fine mesh. Such a system has been developed

by *AtSeaNova* building on the R&D work of the EU AT-SEA project as shown in Figure 18. Using a standard $2 \times 10$ m fabric module yields of up to 14 kg/m of *Saccharina latissima* have been reported off the Irish coast with automated machinery to harvest sheets and reseed them through a spray-binding process [23].

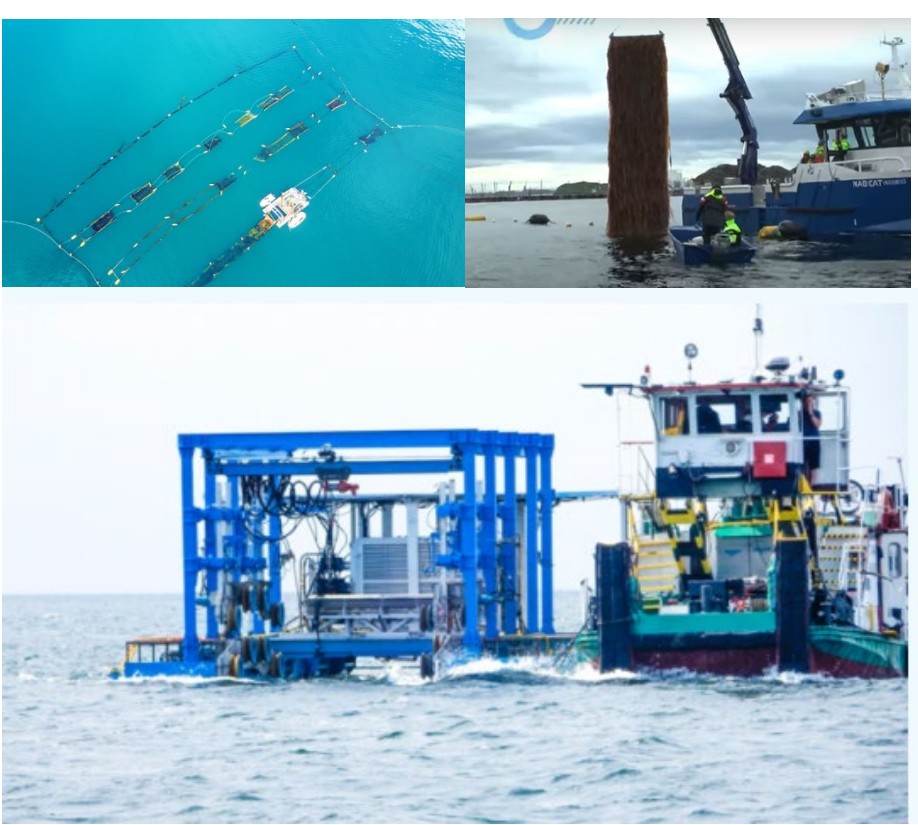

**Figure 18.** The 2D Fabric substrate system and harvesting machine—images—https://atseanova.com/ (accessed on 22 September 2022).

A second 2D system making use of vertical sheet-like nets allowed to float freely from a single point mooring has been patented by *Seaweed Energy Solutions AS* (as shown in Figure 19) [15]. There is scarce published information on how such systems can be efficiently harvested and reseeded.

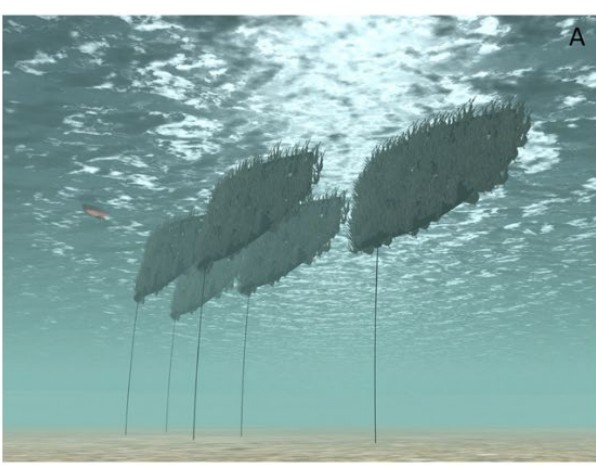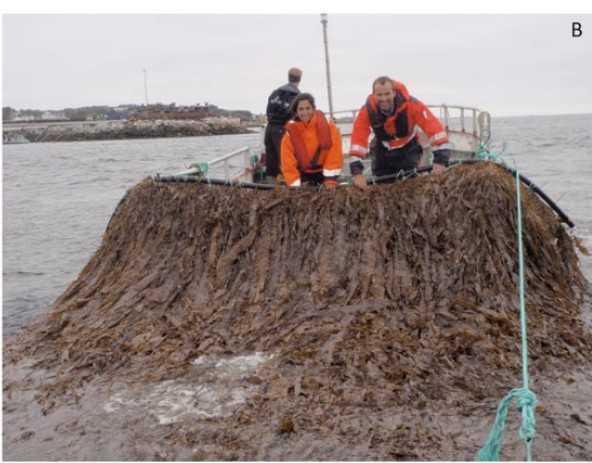

**Figure 19.** The Seaweed Carrier–*Seaweed Energy Solutions* AS—using a sheet-like net with single cable mooring—system can be expanded with up to 20 sheet carriers on longline system per mooring. (**A**) Concept, (**B**) net at harvest, *Reproduced with Permission* [15].

More recently, a new design of linear net system has been tested in the North Sea off the coast of the Netherlands. This *Cultivator* system, as shown in Figure 20, has claimed to withstand storm conditions with wind gusts up to 124 km/h. The developers are still optimising the system to retain seaweed in storm conditions, and it is not clear as yet how the system will allow for highly efficient harvesting.

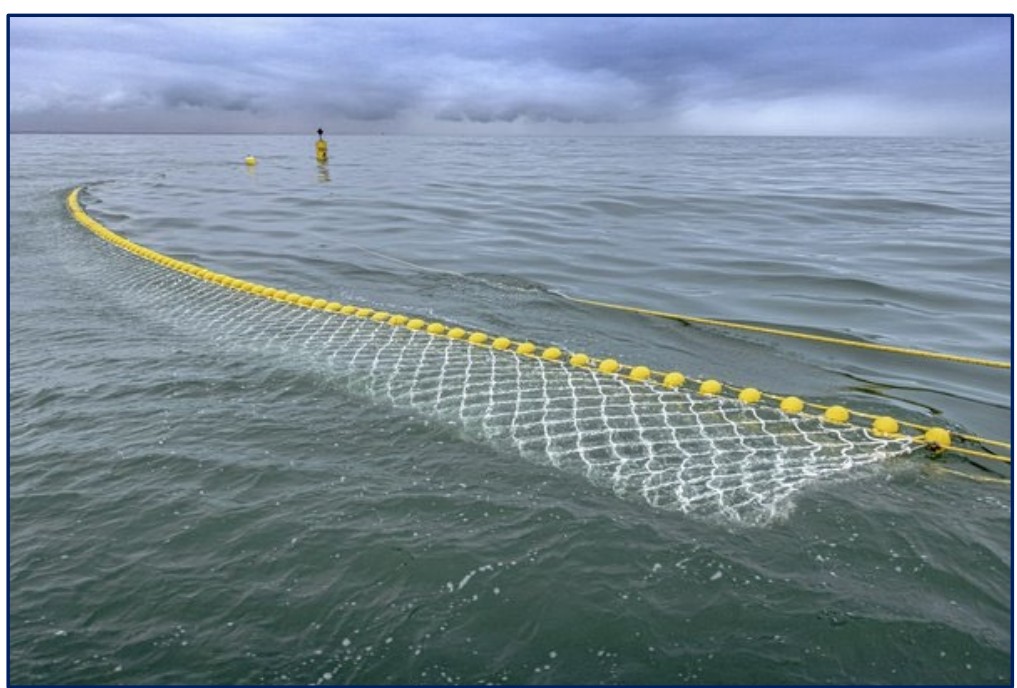

**Figure 20.** "Cultivator" system under testing in North Sea, https://www.northseafarmers.org/news/210518updateots (accessed on 22 September 2022).

## 4. Co-Cultivation of Seaweed and Other Species

Coastal waters adjacent to intensive primary agriculture present the opportunity to both reduce the problem of eutrophication (dead zones) caused by riverine runoff of fertilisers and sewage, and supply nutrients (primarily nitrates, nitrites, and phosphorus) for seaweed farming. It has been estimated that there are some 400 dead-zone systems worldwide affecting nearly 250,000 km$^2$ of ocean that could benefit from a more holistic approach incorporating seaweed farming [39,40]. Alternatively, the bio-extractive nature of seaweed growth can be exploited to feed off the excess feed and faeces generated by intensive fish farming which is itself under pressure to move into deeper offshore waters owing to environmental concerns. This concept can then be taken a stage further with synergistic IMTA systems [41]. A nearshore exposed IMTA scheme proposed in Hawaii aims to harvest seaweed in close proximity to intensive farming of herbivorous fish species (e.g., Rudderfish) that feed on the harvested seaweed, and in turn supply faecal nutrients to the seaweed thus reducing the excess nutrient load in surrounding waters [9].

Co-cultivating seaweed with other marine species (such as fish, prawns, mussels, oysters, and sea cucumbers) is recognised as a promising solution to increase the sustainability and profitability of aquaculture [14,42]. In such an IMTA system, standard cultivation infrastructure for seaweed (discussed in Sections 2 and 3) can still be utilised by simply placing seaweed farms next to other aquaculture farms. An example of such a side-by-side arrangement is the 100 km$^2$ IMTA system in Sanggou Bay (China), as shown in Figure 21. The high-level species inter-relationships are reproduced in Figure 22 [43]. Cultivation infrastructure for some species (such as mussels, scallops, oysters, and clams) may also be integrated with the seaweed cultivation infrastructure as seen in the illustration in Figure 23 [44]. In this integrated design, the longlines and buoys need to be designed to accommodate the loading from oyster/mussel/scallop/clam cages and lines, in addition

to the loading from seaweed lines in waves and currents. As compared to the side-by-side arrangement, this integrated design seems to require less ocean space. However, it is more difficult for operations including harvesting species having different growth times, where harvesting operations for seaweeds may damage oyster/mussel cages and lines.

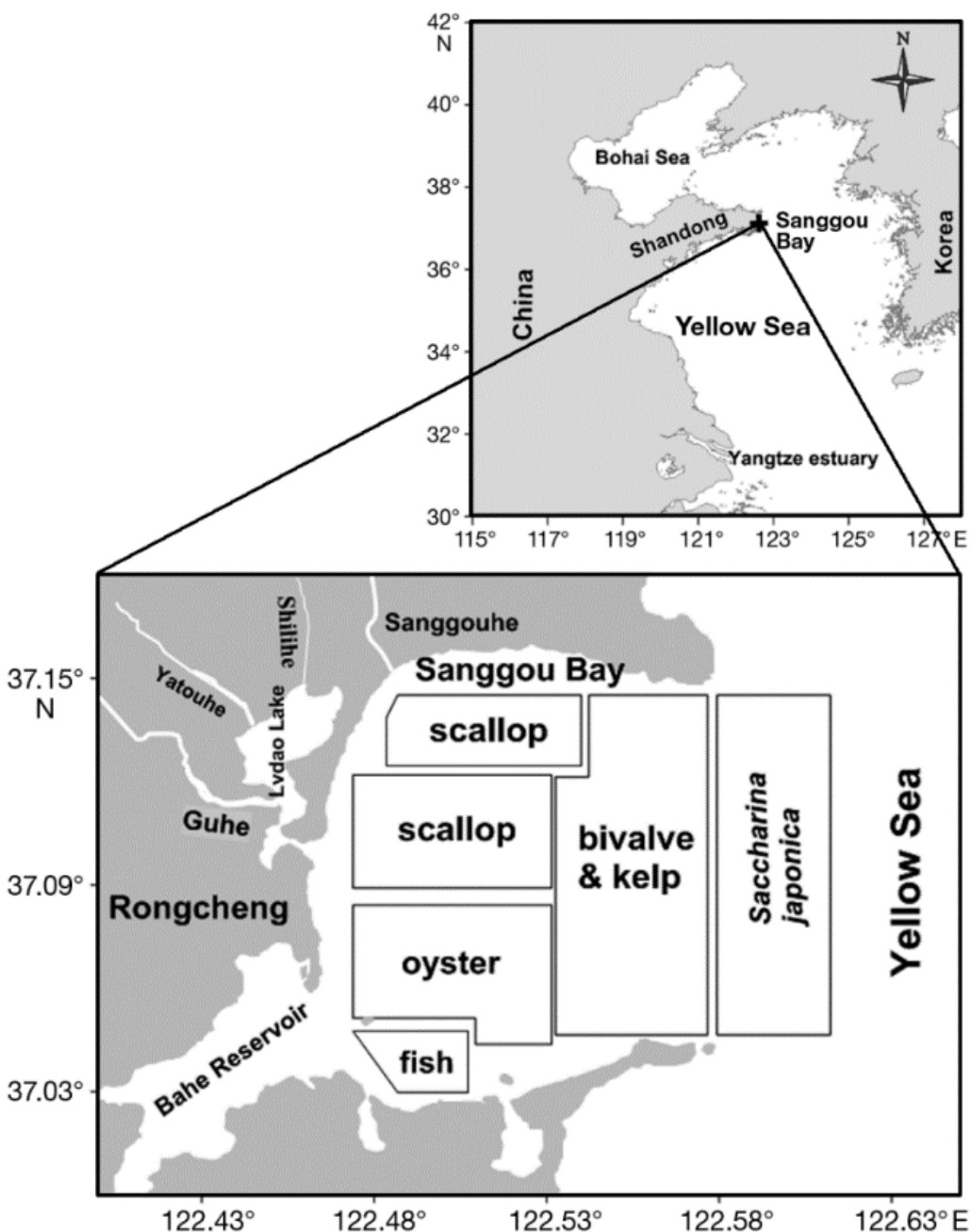

**Figure 21.** Integrated multi-trophic aquaculture system in Sanggou Bay, China [43].

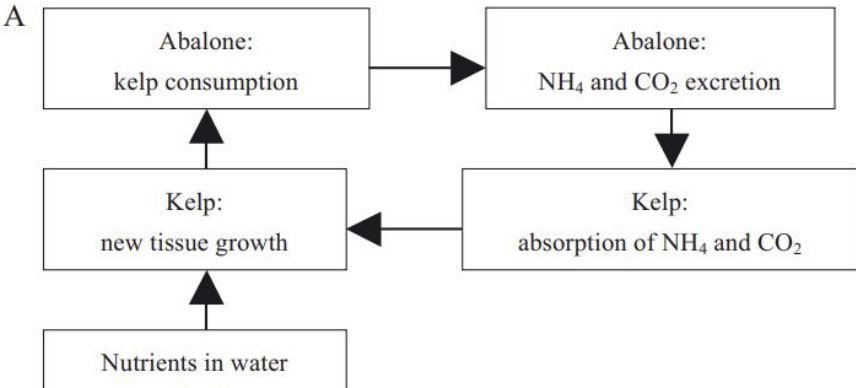

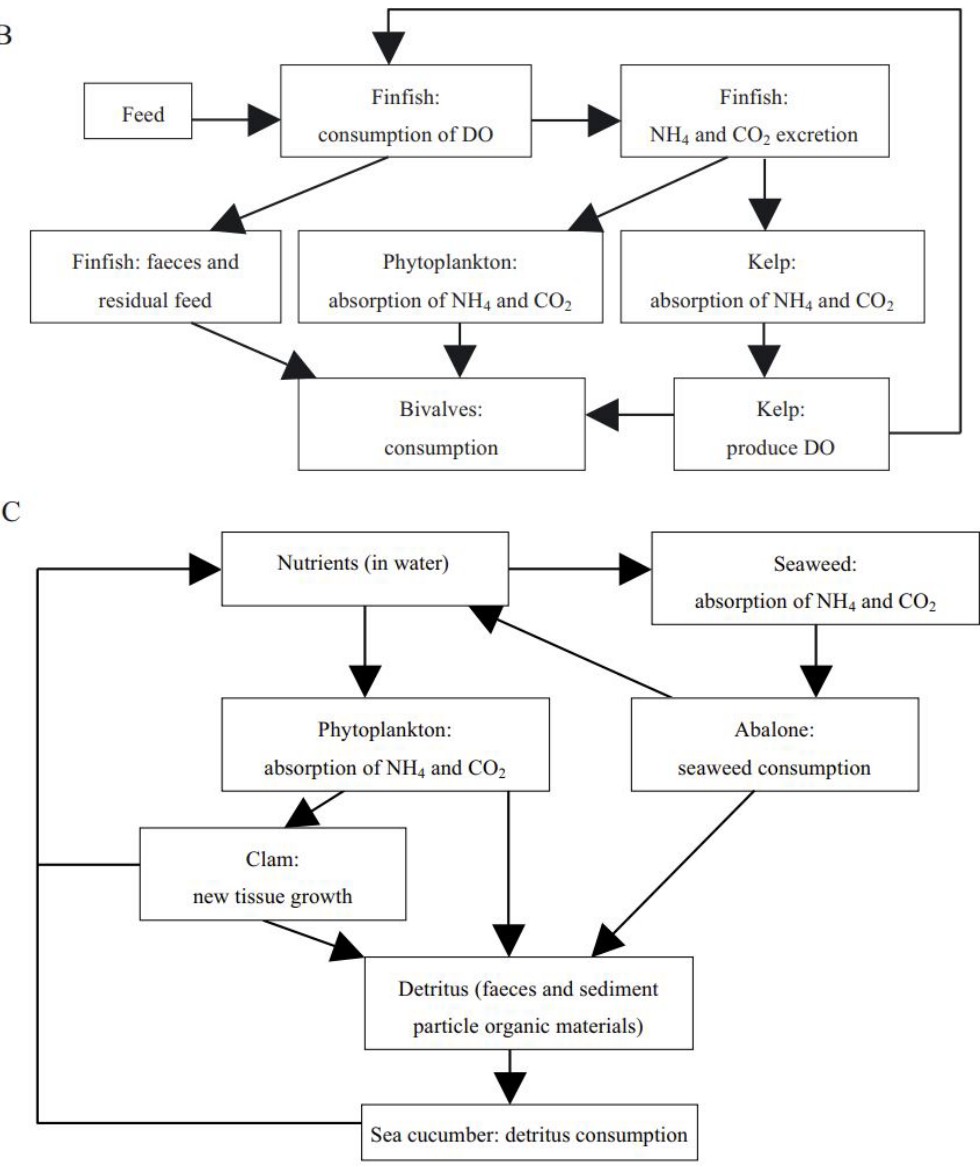

**Figure 22.** IMTA modes in Sanggou Bay, China. (**A**) Long-line culture of abalone and kelp, (**B**) long-line culture of finfish, bivalve, and kelp, and (**C**) benthic culture of abalone, sea cucumber, clam, and seaweed (DO—dissolved oxygen) [43].

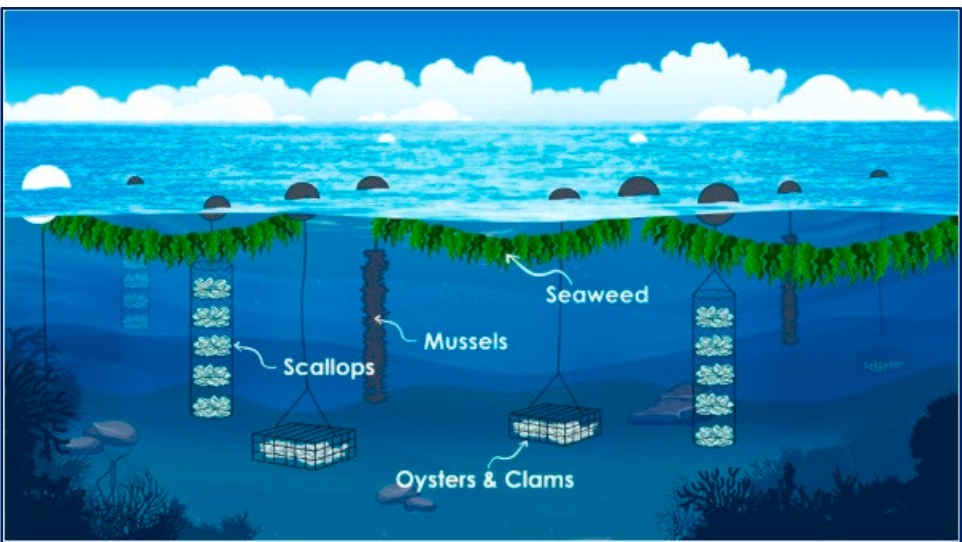

**Figure 23.** Shared farming Infrastructure. *Reproduced with permission* [44].

Trials conducted off Scotland's west coast compared the performance of two seaweed species grown adjacent and remote from salmon farms. The availability of excess nutrients added via fish feeds and faeces showed the benefit of close colocation. The growth rates of these two species (*Palmaria Palmata* and *Saccharina latissimi)* were enhanced by 48% and 61% respectively, with biomass yield improvements of 63% and 27% [45].

This trial reported enhanced ammonia levels in waters up to 200 m from the salmon farms. Other European studies have shown elevated levels of nutrients due to fish farming at the scale of 8 km$^2$ [46] and within 3–5 km [47]. It is clear that the design, layout, and feature scale of future IMTA systems will need to be optimised for the growth of the included species whilst maintaining sufficient water quality (bioremediation) and providing for efficient operational access for harvesting and service vessels.

## 5. Colocation of Seaweed/Other Species and Offshore Renewable Energy Farms

Given the expected rapid uptake of offshore renewable energy expected in many coastal regions a natural extension of this principle is to co-locate aquaculture (Seaweed/IMTA—other species) with offshore renewable energy. Global wave power within 50 km from the coast has been estimated at about 2.11 TW, which if fully utilised could provide 18,500 TWh (almost equal to the 2009 global electricity consumption) [48,49]. The total offshore wind power potential has been estimated to be about 15 TW for water depths up to 200 m, and 5.5 TW for water depths up to 50 m [50]. Currently, the lowest overall cost of offshore renewable technology is wind power [51] albeit fixed bottom installations in depths up to 60 m are cheaper than rapidly developing floating offshore wind power currently.

The rapid growth of offshore wind power [52] makes the usage of the space between turbines more attractive for aquaculture farming as the space is generally too tight to allow for coastal shipping. Co-located aquaculture thus adds value to the leased area. These leases could include other technologies such as wave power-buoy systems such as those developed by *Carnegie Clean energy* and Sweden's *CorPower*, floating offshore solar, and seaweed/IMTA aquaculture. The development of seaweed/IMTA aquaculture and offshore wind co-location is being supported by the Weir and Wind project and UNITED project in the North sea. [53,54]. A recent investigation of stakeholders in Germany showed that most of them supported in principle the co-location and integration approach, but a range of technological, biological, economic, and legal constraints was identified [55]. Key technological concerns include: (i) high costs for development and application; (ii) safety for workers; (iii) impacts on operational activities for wind farms; and (iv) stability/robustness of wind turbines connected with seaweed cages/lines (dual-use moorings).

Several concepts have been advanced previously, some of which make use of the base of wind turbines as a mooring as shown in Figure 24 [56]. These concepts only allow for small-scale offshore farming of seaweed and mussels and are inherently more difficult to automate for harvesting and reseeding.

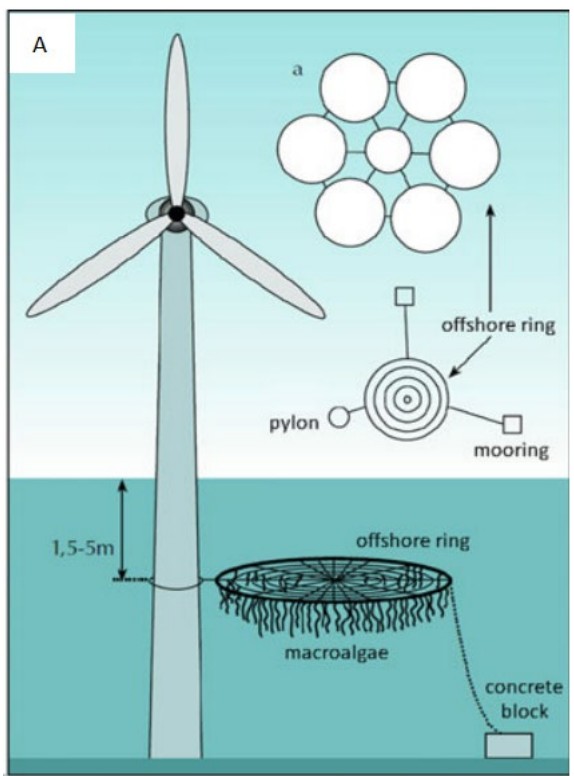
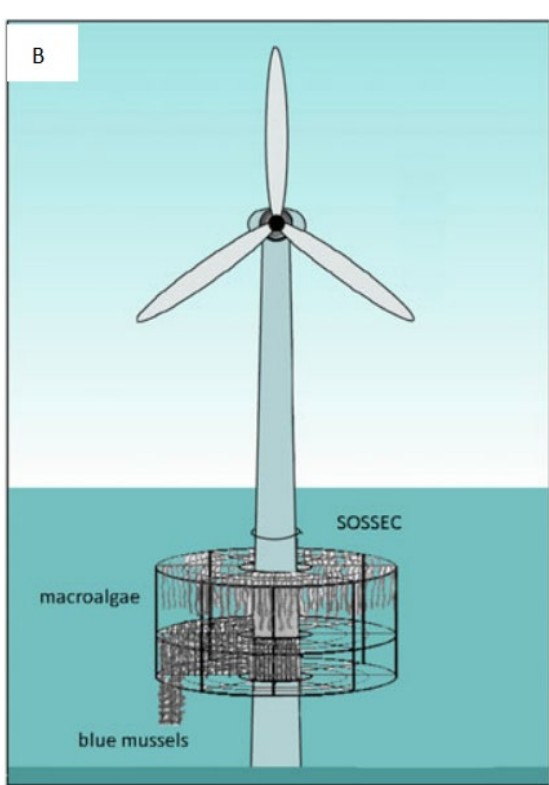

**Figure 24.** Integrated aquaculture—wind turbine. (**A**) Ring system, (**B**) Integrated mussel-seaweed system [56].

Another concept envisages separately moored cable tension systems such as the BAL system within offshore wind parks as shown in Figure 25 [13]. These systems are showing the greatest promise for increased productivity through mechanization as described in Section 4. Similarly, mussels have been successfully cultivated on longline systems with increasing levels of mechanization in more exposed waters in New Zealand albeit incurring higher maintenance costs than nearshore sites [57]. In more energetic water surfaces, floatation buoys transfer energy to the submerged structure leading to potential product loss and increased wear. Structures that keep the vast bulk of their buoyancy elements below worst-case wave turbulence are likely to be subject to much lower extreme case loading and more efficiently use the buoyancy element volume. Surface buoys then become primarily tall visual locators that impart minimal energy. Such a novel system (as shown in Figure 26) intended for offshore mussel farming is under testing in New Zealand's Bay of Plenty [57].

It is easy to imagine how such a system could be blended with the BAL culture line matrix, and linear automated harvesting and reseeding to produce a highly efficient, large-scale, relatively lightweight structure for seaweed farming that is able to resist extreme forces. The previously developed air-filled *Seastrut*$^{TM}$ beams could provide a floatation element that would survive extreme wave or storm forces. A key consideration for offshore wind park developers in working with the aquaculture sector is contingency planning for the case in which an aquaculture structure becomes unmoored and wraps itself around a turbine foundation [56]. Flexible, lightweight systems will deform presenting minimal frontal area and drag whereas a submerged platform with internal compressive strength is less likely to deform and hence lead to much worse consequences.

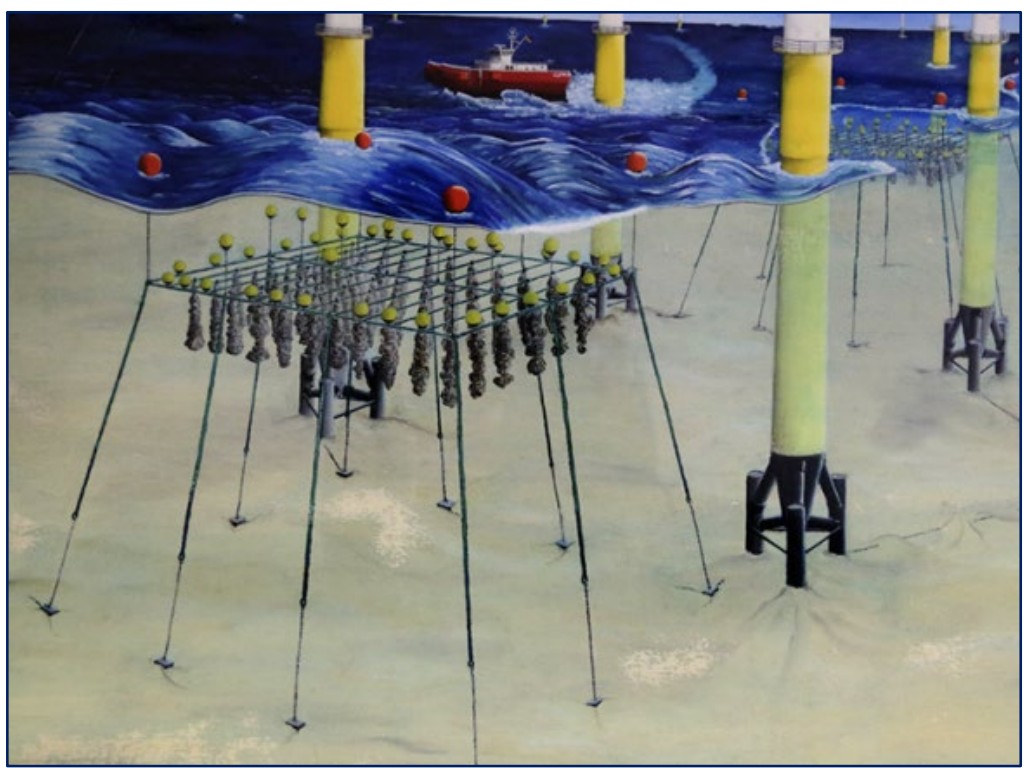

**Figure 25.** Cable tethered seaweed/IMTA–tensile system in offshore wind park [13].

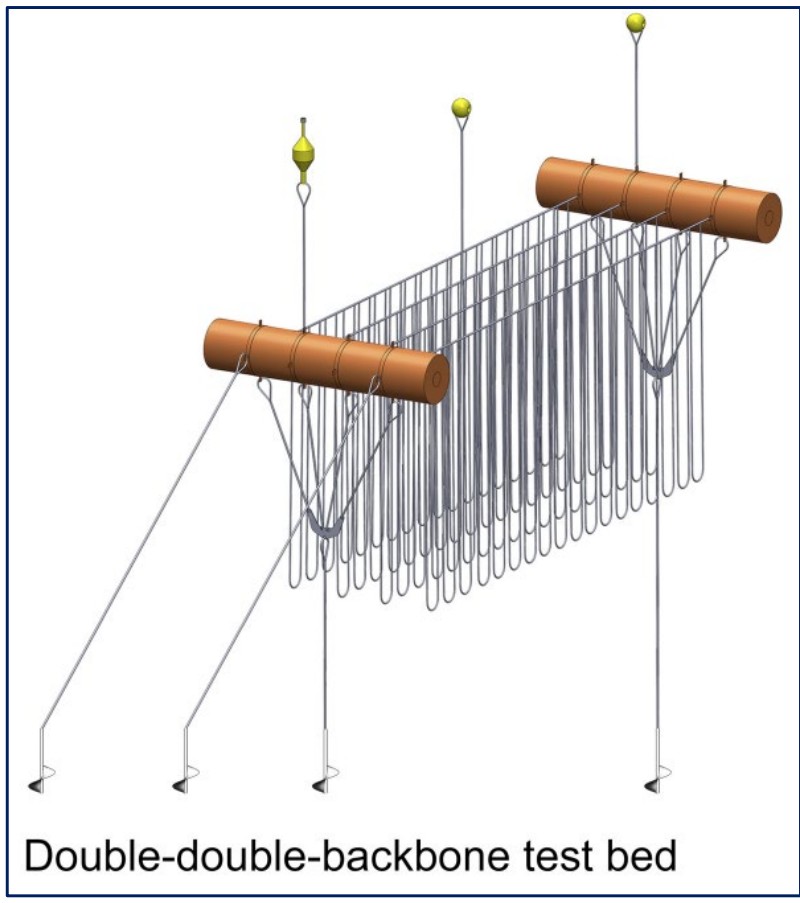

**Figure 26.** Submersible mussel cultivation structure [57].

In addition to the value added to an offshore wind farm site by aquaculture, it has been estimated that the synergies of co-location - shared labour and support infrastructure could lead to a 10% reduction in farm operating and maintenance (O&M) costs [56]. One possible option is to utilise shared vessels for wind turbine maintenance access and aquaculture long-line harvesting, reseeding, and inspection. The use of active motion stabilising systems onboard such vessels allows safe crew operation in larger significant wave heights thereby increasing the service availability (% time) for aquaculture or wind turbine maintenance. The reduction in "weather downtime" leads to a reduction in turbine outage time thus reducing O&M costs of the wind farm operation.

The siting of offshore wind farms (driven primarily by the requirement for high average wind speeds and a preference for waters less than 60 m depth to minimize foundation costs) is fortuitously consistent with many of the potential siting requirements for exposed waters aquaculture. Excessive depth for the siting of fed finfish stock is likely to lead to the plume of excess feed and faeces falling to the seafloor and mixing less with sub-surface aquaculture crops. Both aquaculture and offshore renewables will have lower operational costs the closer they are to harbour ports. In exposed (not protected) coastal locations there is likely to be a minimum depth requirement for aquaculture because larger storm waves carry higher lateral energy as they move into shallower water [57].

For IMTA system's incorporating finfish farming (e.g., salmon) the feed supply is typically delivered by diesel powered feed barges both for propulsion and feed pumping. If an IMTA cultivation system is able to supply and exchange nutrients based on the excess nutrients of finfish farming, the next logical extension of a co-located IMTA/seaweed and coastal offshore energy system is to decarbonise the feed barge operation. One possible solution under development is based on the supply of wave power to power the feed pumping as envisaged in the Moorpower$^{TM}$ floating barge wave power system as shown in Figure 27.

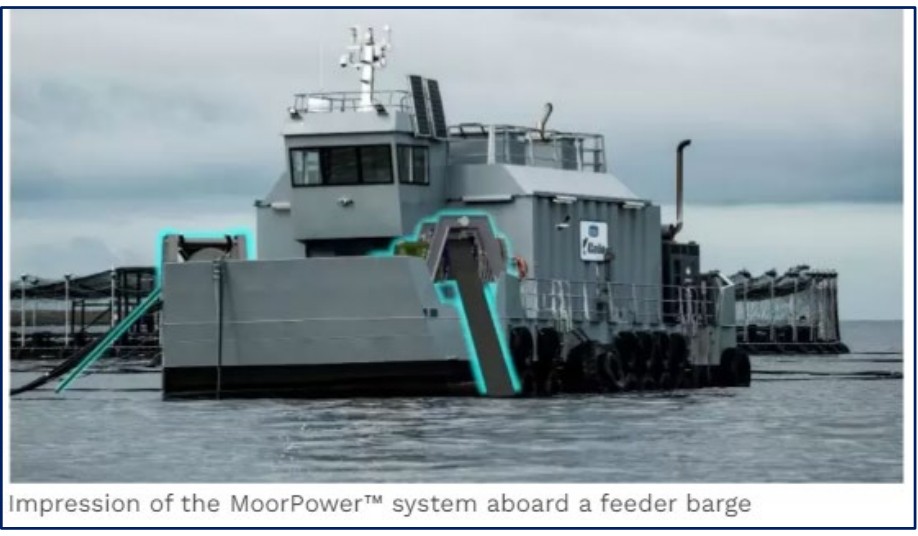

Impression of the MoorPower™ system aboard a feeder barge

**Figure 27.** Moorpower$^{TM}$ floating feed barge showing wave energy converters in green—image from https://blueeconomycrc.com.au/project/moorpower-scaled-demonstrator/ (accessed on 22 September 2022).

An alternative concept to such a system would use an electrical offtake from an offshore wind (or wave) farm to supply electrical power to a feed barge via an electrical power take-off buoy doubling as a mooring solution for a feed barge. A grid-connected offshore wind or wave buoy farm would be able to supply energy in the absence of wind or waves. These preliminary concepts are illustrated in Figures 28 and 29. In the case of a wave buoy matrix the wave energy extraction would help to shelter the aquaculture downstream. Taking this one step further it may be possible to surround an offshore IMTA system that exists within an offshore wind farm with an annular wave buoy matrix as

shown in Figure 30. This would yield both additional grid-connected power and wave attenuation leading to enhanced operational safety and increased availability of access. Realisation of such a concept would require a significant improvement in the levelised cost of energy (LCOE) of wave buoy power. Unfortunately, most wave buoy systems currently close down for self-protection in storm conditions reducing this benefit in the extreme case.

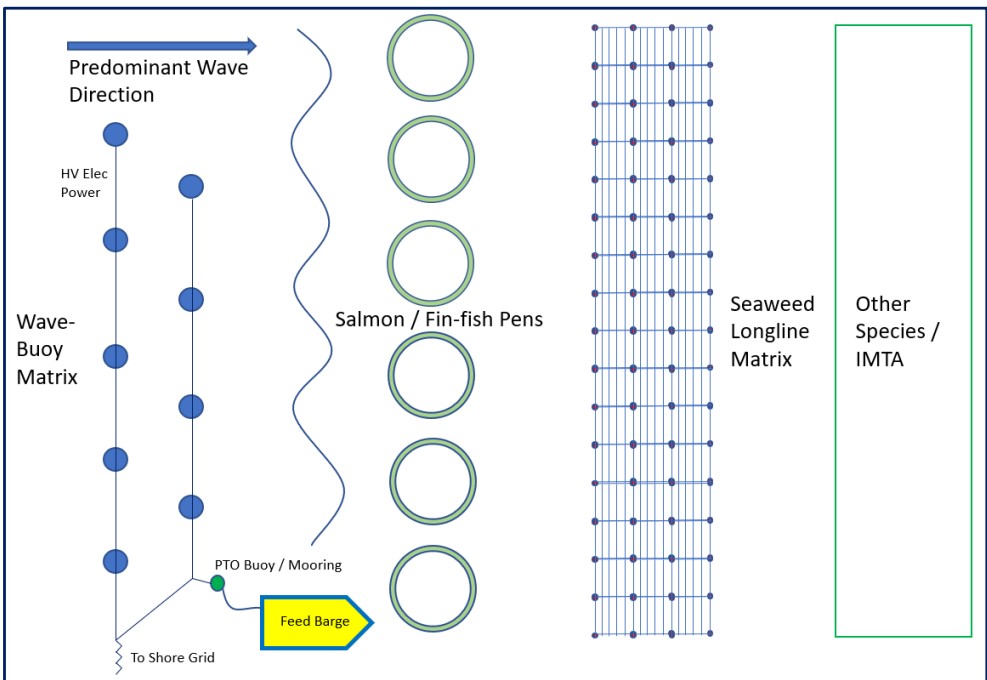

**Figure 28.** Grid connected-wave buoy powered feed barge, IMTA seaweed system concept.

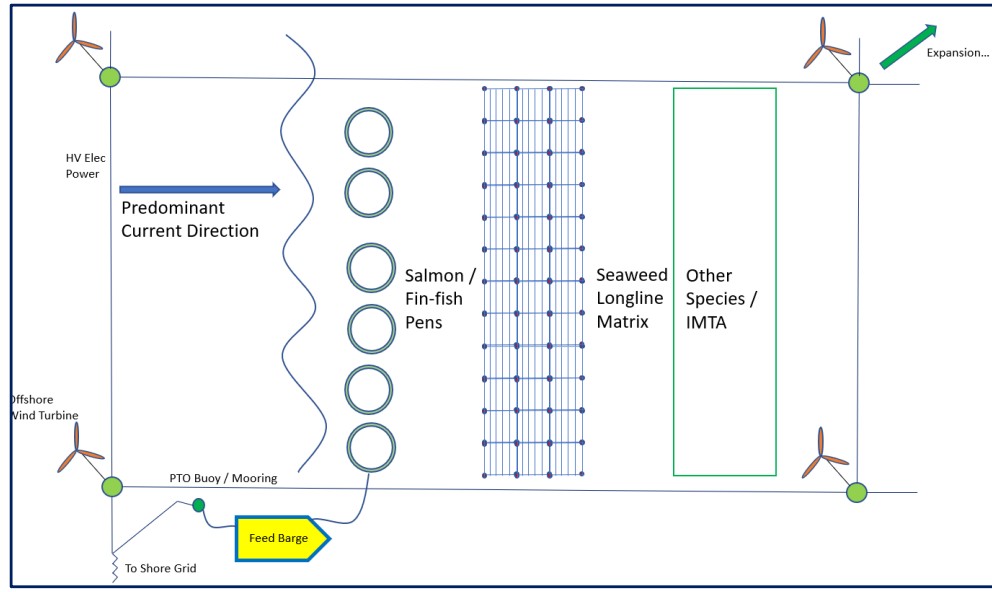

**Figure 29.** Grid connected offshore wind powered—IMTA, seaweed system concept.

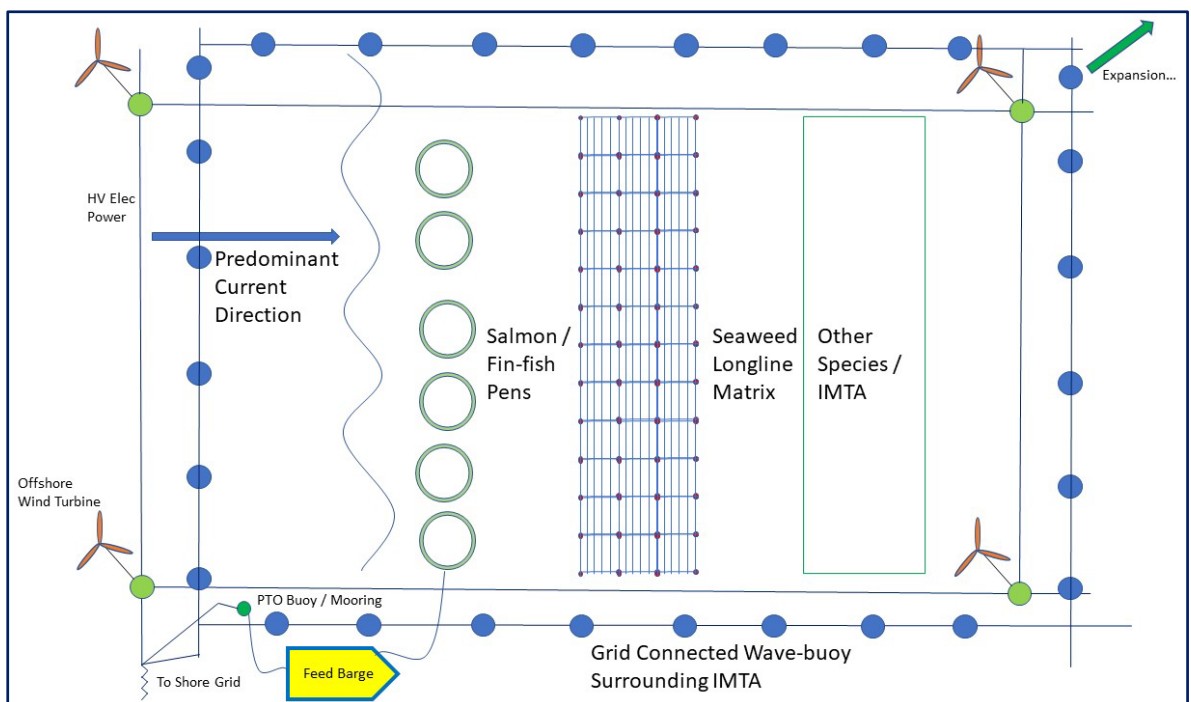

**Figure 30.** IMTA system surrounded by wave suppressing grid connected wave buoy annulus.

## 6. Conclusions

In developing regions such as India, Bangladesh, and much of Africa which are expected to see considerable population growth the development of large-scale, low-cost coastal aquaculture (seaweed and seafood) offers a highly sustainable solution to impending food challenges that avoids the need for limited freshwater, fossil-fuel powered fertiliser production, and arable land. For these regions the development of nearshore IMTA systems pioneered in East Asia offer an affordable pathway to increasing food production.

The expansion of seaweed cultivation into the broader world community beyond East Asia offers both challenges and opportunities in high labour cost countries. As a predominantly low-value crop the economic case for adoption can be greatly improved by combining cultivation systems with other higher-value species (IMTA) and developing highly efficient, low-cost production systems. In many cases adding seaweed cultivation in close proximity to existing nearshore-sheltered finfish farming has been shown to increase seaweed yields and reduce the negative water quality impacts of intensive fish farming.

Alternatively, moving to more exposed waters provides for an order of magnitude expansion of co-located seaweed and seafood aquaculture. This can then be further improved with the best synergistic combination of grid-connected offshore renewable energy systems such as offshore wind power and perhaps offshore wave buoy power in powering aquaculture feed barges. The full exploitation of offshore wind farm areas with co-located offshore IMTA aquaculture both increases the return on the combined marine lease and reduces the O&M costs of offshore wind farms through the use of shared service vessels, port infrastructure, and maintenance labour. Although wave buoy power is at an earlier stage of development, with sufficient learning and cost reduction, it could be used to surround an offshore IMTA system that might itself exist within an offshore wind farm. This would provide additional wave attenuation within the IMTA and thus improve operator safety and farm access availability.

It is clear from many trials around the world that seaweed and other farm structures (e.g., bivalves) will need to be submerged to avoid and survive high energy waves in storm conditions. Combinations of the BAL tension pegged system with innovations such as mechanized culture line trimming/harvesting and improved floatation design may well improve cultivation productivity sufficiently to overcome any cost increases depending

on the seaweed species under consideration. Cultivation yields and productivity are likely to be enhanced by both advances in the scientific and operational layout of IMTA systems and the accumulation of many small contributing advancements. For example, with the BAL system, culture lines are currently removed and reattached by divers during harvesting—this could be mechanised with a simple tool on a pole.

Step change solutions requiring high levels of technology automation such as the *Gili SUBFlex* depth-cycling concept, inflated flexible beams, or renewably powered nutrient upwelling systems are likely to introduce a large increase in system cost. Such systems are likely to have higher maintenance costs and be inherently less reliable than well designed passive cultivation systems. Early experimental trials have shown that technology can be used to overcome the growth limiting lack of surface water nutrients and excessive water temperature in exposed water. However preliminary analysis suggests these are unlikely to be economically feasible for seaweed farming alone but may be worth further investigation when used in combination with higher value finfish or bivalve aquaculture.

**Author Contributions:**

| Contributor | Contribution Type | Percentage (%) |
| --- | --- | --- |
| R.M.T. | Conception, discussion, conclusions | 70 |
| | Literature research | 60 |
| | Review, checking | 20 |
| H.P.N. | Conception, discussion, conclusions | 20 |
| | Literature research | 30 |
| | Review, checking | 30 |
| C.M.W. | Conception, discussion, conclusions | 10 |
| | Literature research | 10 |
| | Review, checking | 50 |

All authors have read and agreed to the published version of the manuscript.

**Funding:** The authors acknowledge the financial support of the blue economy Cooperative Research Centre (CRC), established and supported under the Australian Government's CRC Program, grant number CRC-20180101. The CRC program supports industry-led collaborations between industries, researchers, and the community.

**Institutional Review Board Statement:** Not applicable.

**Informed Consent Statement:** Not applicable.

**Data Availability Statement:** Not applicable.

**Conflicts of Interest:** The authors declare that they have no potential conflicts of interest in publishing this literature review.

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
