# Peer review of "Review of the Status and Developments in Seaweed Farming Infrastructure"

_jmse, doi:10.3390/jmse10101447_

Round 1
Reviewer 1 Report
Seaweed farming has been highly valued around the world for its promising potential to provide the decarbonization and reduce water eutrophication. This article overviewed traditional and modern seaweed farming infrastructure, provided elaborated value assessment on recent innovations in seaweed cultivation, gave suggestions on seaweed growth acceleration. This is arduous and meaningful work that reflects the author's hard work.
The paper is well-written. Methods are well explained and fit with the objectives proposed. Discussion and conclusions need to be more in-depth. The language needs to be streamlined.
The following are my suggestions:
1. Lines 53-56, The sentence is quite long and is not straightforward enough. Can the authors simplify this sentence, for example summarize the whole paragraph with a 2-lines sentence? This may make the paper much more readable.
The suggestion of simplifying sentences is suitable for the whole paper's content.
2. Table 1 and Table 2, adjust the table layout, try to put the table on one page.
The format of table 2 cells are inconformity.
3. The line number is discontinuous on page 6 (although it’s not a problem regards on publishing, it’s a problem for reviewing).
4. After every figure title “fig. x”, there is a line “-”, the lines are inconformity among titles.
5. the figures mentioned in the text didn’t show up according to the number order, for instance, fig.3; fig 9, fig 22…
6. fig.23, definition of this picture should be improved.
7. The authors provided us with the concept of synergistic combination of offshore renewable energy systems and sea farming. This is a very new cross-directional marine engineering, which has received extensive attention. The authors can make more in-depth comments on the impact of this joint project on the economics and safety of sea farming, which readers might be concerned about.
Author Response
Thanks for your review. I've attached a reply to your points.

Reviewer 2 Report
- The authors should revise the title to clearly announce this a review paper.
- Explain what do you mean by infrastructure in specific ways. What items?
- L16-18: rephrase the sentence, please
- L25, L77, L121 in P. 10, Fig. 7 caption, Fig. 10 Fig. 7 caption, L41 in P.16, L139 in P.21 etc: Explain the acronyms fully for the first time. Please check it throughout the manuscript.
- L31: I reckon that the new data is available for 2020.
- L52: Please add “.. as nursery grounds”
- L53-62: This paragraph about beef cattle is placed in the introduction section without connection with the previous and subsequent paragraphs. Therefore, it is redundant and should be deleted.
- L117: what do you mean by “seedling propagule” is it the main or branched seedling line?
- L128: why the authors italic the word “most”?
- L130: It is not common to italic sp. or spp.
- Table 2 , L132-134: It is suggested to mention the appropriate depths of seaweeds for their cultivation practices.
- L 57 in P.9: consider the sea transportation issues in the challenging systems.
- L78 in P. 10: only italic the scientific names.
- Fig. 5: It is blurry maybe need to redraw.
- Table 3: The authors used citation in the caption. It means that the whole table is adapted from reference no. 15 without any changes? If yes, the authors should remove the table from their manuscript and only describe the highlights of it in the text, then readers who want know detailed information can browse the reference no. 15.
- Fig 8. Is it better to say “H-frame” or “H-shaped frame”?
- Fig. 8 A: The text inside the figure is not readable even with high magnification
- L34 in P.16: what do you mean by “SUBFlex”? Is it a commercial name?
- Fig. 13: I think this picture adopted from a video, because the bottom of the image has a line to show the duration of the video. Please crop it.
- L165 in P. 22 or L223 in P.26: Please explain even with pictures to show how seaweeds play role in integrated fish cage culture to have a self-sustaining cage culture.
- This manuscript is fulfill with the seaweed rearing merits, however, the authors should consider some disadvantages of seaweeds artificial culture from open to near shore sea areas in terms of possible side-effects on changing fish and aquatics animals migration routes, side-effects on pelagic aquatics or bottom dwellers populations according to the different rearing techniques, changing the balance of natural nutrients in the cultural regions etc. Maybe creating a section is needed.
Author Response
Thanks for your review. I've attached a line by line response.

Round 2
Reviewer 1 Report
The language of this article has been greatly improved, and the content and discussion are also improved.
But the layout of the article is still messy, especially the pictures and tables, the clarity is also low, which will affect the reading and understanding of the article.
The authors please deal with this problem, after this I think the article can be accepted.
Author Response
I'm glad you agree that the language, discussions, and conclusions are much improved - this is the most important thing.
I have improved the layout and figure captions so they are all consistent. I have also checked that all the images meet the minimum clarity requirements - all text is clearly readable with some magnification for the smaller text.
I trust that now meets all your concerns. Thanks for your reviewing efforts.
Reviewer 2 Report
The authors carefully addressed the comments. However, there are some minor flaws as a follow:
According to the authors’ responses, please revise the title.
According to the authors’ responses, I know that all environmental conditions are important for growing algae, however, provide the optimal depth as many as possible (if it is possible).
According to the authors’ responses, it is suggested (if it is possible) to show the system of algae production in an integrated fish cage culture as a sustainable method.
Author Response
I have revised the title as you suggested to clearly label it as a review - even though it includes new material and concepts.
We have decided that we cannot access enough accurate information on the species listed to comment on their ideal growing depths as there is both a lot of variability by location and scarce information.
I still do not understand what you have meant by your last point and you haven't added any further clarification so we cannot comment any further. The paper includes copious discussion of integrated IMTA systems, potential systems with herbivorous fish, and discussion of the infrastructure. I trust you are happy to accept that.
Many thanks for your reviewing efforts.